# Microglial expression of CD83 governs cellular activation and restrains neuroinflammation in experimental autoimmune encephalomyelitis

Pia Sinner [1], Katrin Peckert-Maier [1], Hashem Mohammadian [2], Christine Kuhnt[1], Christina Draßner[1], Vasiliki Panagiotakopoulou [3,4], Simon Rauber [2], Mathias Linnerbauer[5], Zhana Haimon[6], Dmytro Royzman[1], Deborah Kronenberg-Versteeg [3,4], Andreas Ramming[2], Alexander Steinkasserer[1] & Andreas B. Wild [1] ✉

Microglial activation during neuroinflammation is crucial for coordinating the immune response against neuronal tissue, and the initial response of microglia determines the severity of neuro-inflammatory diseases. The CD83 molecule has been recently shown to modulate the activation status of dendritic cells and macrophages. Although the expression of CD83 is associated with early microglia activation in various disease settings, its functional relevance for microglial biology has been elusive. Here, we describe a thorough assessment of CD83 regulation in microglia and show that CD83 expression in murine microglia is not only associated with cellular activation but also with pro-resolving functions. Using single-cell RNA-sequencing, we reveal that conditional deletion of CD83 results in an over-activated state during neuroinflammation in the experimental autoimmune encephalomyelitis model. Subsequently, CD83-deficient microglia recruit more pathogenic immune cells to the central nervous system, deteriorating resolving mechanisms and exacerbating the disease. Thus, CD83 in murine microglia orchestrates cellular activation and, consequently, also the resolution of neuroinflammation.

Microglia, the tissue-resident macrophages of the central nervous system (CNS) parenchyma, are vital for maintaining tissue integrity and homeostasis[1]. They are not only involved in the clearance of cellular debris or pathogens but also shape synaptic plasticity and promote myelination of axons during development[2–4]. Their homeostatic phenotype and functions depend on instructive signals from the environment, such as TGF-β or CSF1R-ligands[5–7], and accordingly,

elaborated single-cell transcriptomic analyses revealed that such environmental cues cause a pronounced regional heterogeneity in microglia throughout the CNS[8,9].

The prominent role of microglia in tissue maintenance and their dependence on intact signaling from the surrounding predestine them to react rapidly to tissue alterations. For instance, neurodegeneration or neuroinflammation elicits a gradual progression from homeostatic

---

[1]Department of Immune Modulation, Uniklinikum Erlangen, Friedrich-Alexander Universität Erlangen-Nürnberg, 91052 Erlangen, Germany. [2]Department of Internal Medicine 3, Uniklinikum Erlangen, Friedrich-Alexander Universität Erlangen-Nürnberg, 91054 Erlangen, Germany. [3]Department of Cellular Neurology, Hertie-Institute for Clinical Brain Research, University of Tübingen, Tübingen 72076, Germany. [4]German Center for Neurodegenerative Diseases (DZNE), Tübingen 72076, Germany. [5]Department of Neurology, Uniklinikum Erlangen, Friedrich-Alexander Universität Erlangen-Nürnberg, 91054 Erlangen, Germany. [6]Departments of Immunology and Regenerative Biology, Weizmann Institute of Science, Rehovot 76100, Israel. ✉e-mail: andreas.wild@uk-erlangen.de

to disease-induced gene expression, also known as disease-associated microglia (DAM)[10–12]. Due to the transitional nature of these adaptations to potential damage, factors that could modulate a potentially harmful disease-associated state would represent promising candidates for intervening in disease progression.

Interestingly, there is a conceivable overlap between the DAM gene signature and expression patterns observed in microglia that are associated with proliferation during CNS development or if CSF1R-signaling is disrupted[13,14], suggesting common regulatory pathways. These datasets comprise several genes related to cellular activation, such as the *Cd83* transcript. The corresponding glycoprotein was originally described as a maturation marker for dendritic cells (DCs) and is predominantly expressed in activated immune cells[15]. Recently, its immunomodulatory properties have become increasingly evident since CD83 expressed by DCs orchestrates their activation status and serves to restrain derailing immune responses[16]. CD83 expression in macrophages is part of an immediate-early response to inflammation[17], and its deletion in those cells causes an impaired pro-resolving phenotype[18]. Thus, CD83 might exert similar regulatory functions during microglial activation. In the CNS, *Cd83* expression is indeed largely confined to homeostatic murine microglia and border-associated macrophages[19,20] and gets upregulated early during disease-associated transitions[11,14,19,21]. Furthermore, *CD83* expression characterizes a specific 'pre-activated' set of human microglia that is reminiscent of DAMs, and it is also more prominent in microglia isolated from the white matter than their gray matter counterparts[12,22,23]. However, despite this body of evidence provided by recent advanced single-cell transcriptomic phenotyping, insights into the importance of CD83 for microglial functions in CNS homeostasis and pathologies are still lacking.

In this work, we present a thorough investigation of the role of CD83 in microglia, both under homeostatic and pathologic conditions. Using CD83eGFP reporter mice, we disclose that in a healthy CNS, CD83 is almost exclusively expressed in microglia although not being uniformly distributed. In the experimental autoimmune encephalitis (EAE) model of neuroinflammation, CD83 expression increases under inflammatory conditions and remains elevated throughout the complete course of the disease. In vitro, we demonstrate that microglial expression of *Cd83* relies on TGF-β and can be strongly induced by IL-4. Targeted deletion of CD83 in microglia reveals a disturbed phenotype and aberrant response to myelin debris. Using the EAE model, we show that specific deletion of CD83 (CD83$^{\Delta MG}$) in microglia results in aggravated autoimmune neuroinflammation, which is marked by a profound influx of monocytic cells into the CNS. Using single-cell RNA sequencing (scRNA-Seq), we disclosed that CD83-deficient microglia display an over-activated phenotype. These cells express higher levels of chemokines and create a TNF-α-driven pro-inflammatory milieu, which causes more severe damage in the brain by higher expression of *Mmp9*. Collectively, we reveal that CD83-deficient microglia are inapt to cope with damage and react with heightened inflammation while showing impaired pro-resolving mechanisms. Our findings provide important insights into the regulation of neuro-inflammatory processes exerted by microglia.

## Results

### Microglial expression of CD83 exhibits a distinctive regional pattern associated with white matter

Since data on microglial CD83 expression is limited, we first sought to compile a detailed and comprehensive description of CD83 expression patterns under physiological and pathological conditions. To this end, we employed reporter animals, which express eGFP under the control of the *Cd83* promoter[24]. In the first approach, we isolated and analyzed immune cells from the CNS of these CD83-eGFP mice to assess the distribution of CD83 expression among various cell types. As expected, reporter activity was detected only in CD45$^+$ cells, i.e., immune

cells, and not in CD45$^-$ cells (Fig. 1a). Of all eGFP$^+$ immune cells, microglia have the greatest share with 93.4% (±1.31%), followed by B cells with 5.8% (±2.57%). T cells, Natural Killer (NK) cells and monocytes make up less than 1% of all CD45$^+$/eGFP$^+$ cells, whereas eGFP expression is virtually absent in neutrophil granulocytes (Fig. 1b). While microglia uniformly express CD83, B cells divide into two populations, one of which expresses high levels of CD83 (Fig. 1c). To rule out that the observed eGFP signal is caused by microglial activation during the isolation process, we isolated microglia in the presence of transcription inhibitor Actinomycin D, which prevents artificial gene expression[25]. In this setting, we did not detect any differences in eGFP-signal strength, confirming microglial *Cd83* promoter activity already under homeostatic conditions (Supplementary Fig. 1b). Since gene expression in microglia is spatially heterogeneous throughout the CNS[12], we further assessed microglial CD83 expression stratified for the brain region from which they originated. While hippocampal and cortical microglia show similar levels of CD83 promoter activity, the eGFP reporter signal increases significantly in further caudal regions and is the highest in the spinal cord (Fig. 1d). In contrast to human tissue, mouse brains mainly consist of gray matter, which contains neuronal cell bodies, and less white matter (e.g., rich in myelinated fibers). Interestingly, regions typically associated with higher content of white matter also express higher levels of CD83. Human RNA data implied an association of *CD83* expression with white matter microglia[23], and thus, we proceeded with immunofluorescent stainings of human brain tissue to corroborate this notion. While staining against CD83 in the human neocortex resulted only in diffuse staining in gray matter, we detected specific signals in white matter, which clearly co-localized with Iba1 staining and thereby were attributed to microglia cells (Fig. 1e). Intriguingly, it appeared that CD83 expression in white matter microglia is associated with a rather rounded cell shape suggesting a different function or phenotype of these cells. Collectively, we demonstrated not only that microglia are the paramount cell type expressing CD83 in the CNS but also that this expression is linked to cellular regions with high myelin content.

### Expression of CD83 in microglia increases with cellular activation and during the resolution of neuroinflammation

Recent single-cell transcriptomic analyses have suggested that CD83 is expressed in "pre-activated" or immune-alert microglial cells in homeostatic microglia and during early activation in a neuro-inflammatory context[11,26,27]. Since microglia rapidly respond when removed from their native environment[5], we first assessed the regulation of CD83 on ex vivo cultivated cells. When we incubated acutely isolated adult microglia for 6 h at 37 °C, we observed a striking induction of surface CD83 expression. By contrast, cells, which were incubated at 4 °C for the same duration, showed no increased staining intensity compared to the background signal of CD83$^{-/-}$ microglia (Fig. 2a). Next, we immunized CD83-eGFP reporter mice with MOG$_{35–55}$ peptide to induce experimental autoimmune encephalomyelitis (EAE), which models the early inflammatory phase of Multiple Sclerosis (MS) and assessed the expression of CD83 in microglia at the peak of disease. We disclosed not only a significant increase in eGFP-reporter activity but also elevated CD83 surface levels in microglia from inflamed CNS (Fig. 2b, c). It has been reported that activated microglia adopt a CD11c$^+$/MHC-II$^+$ phenotype during neuroinflammation[28], and when we examined the phenotype of microglia during neuroinflammation in further detail, we observed that CD83 expression was highest in such activated cells of the spinal cord and the brainstem region, followed by microglia of the cerebellum (Fig. 2d). Additionally, expression of CD83 also increases in activated cortical microglia albeit to a lesser extent than in cells from caudal brain regions (Fig. 2e). These flow cytometry data were further corroborated with two-photon microscopy showing co-localization of eGFP and the microglial marker P2RY12, especially in cells of caudal brain regions (Supplementary

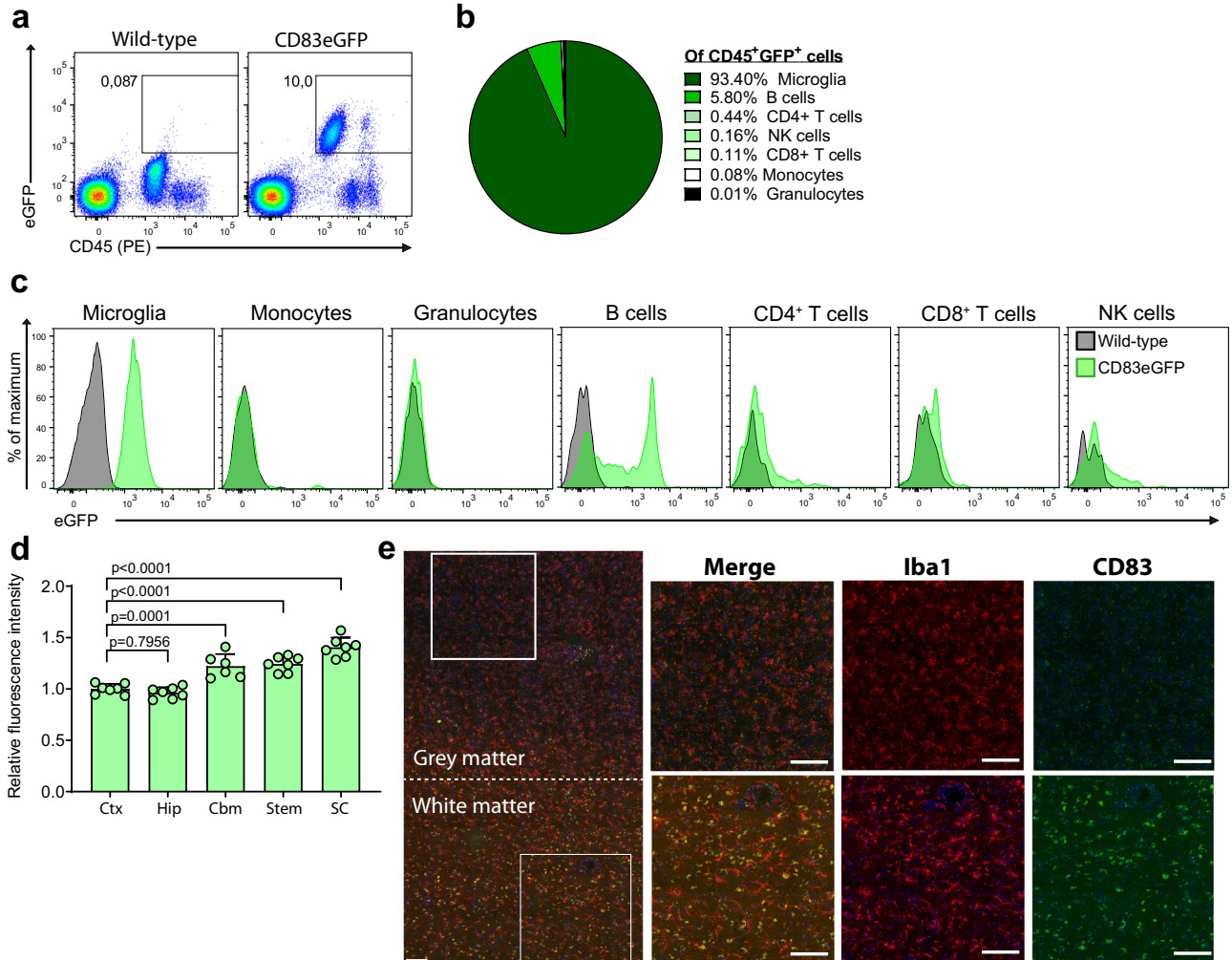

**Fig. 1 | CD83 is expressed in a distinctive regional pattern associated with white matter. a** Flow cytometric assessment of CD83 expression in CD83eGFP reporter animals under homeostatic conditions. Cells were pre-gated on single living cells, and the percentage of eGFP⁺ cells was assessed. Wild-type cells served as negative controls. **b** Summary of the cellular composition among CD45⁺eGFP⁺ cells. Data represent the mean of four different animals. **c** Histograms of eGFP fluorescence in different cell types isolated from the CNS. Wild-type cells served as negative fluorescence controls. Histograms are representative of four individual mice.

**d** Assessment of eGFP signal in microglia from the cortex (Ctx), hippocampus (Hip), cerebellum (Cbm), brainstem (Stem), and spinal cord (SC). Median fluorescence was normalized to cortical microglia ($n = 7$, and each dot represents pooled tissue from two mice). Data are represented as mean ± SEM. Statistical significance was calculated with one-way ANOVA with Dunnett's multiple comparison test. **e** Immunofluorescence of human brain tissue. Microglia are stained for Iba1 (red) and CD83 (green). Nuclear staining was performed with DAPI. Length of all scale bars: 100 µm.

Fig. 1c, d). Interestingly, monocytes, which do not express CD83 under homeostatic conditions, acquired eGFP signal, whereas granulocytes, although massively infiltrating the inflamed brain, do not express elevated levels of CD83 during disease (Supplementary Fig. 1e, f). *Cd83*-promoter activity was exceptionally high in monocytes upon transition to an APC-phenotype (CD11c⁺/MHC-II⁺), which implies that CD83 expression is not exclusively induced in activated microglia during neuroinflammation. However, the eGFP signal in microglia further increases during the recovery phase, especially in the spinal cord, despite declining CD11c expression (Fig. 2f). This indicates a more diverse role of CD83 expression during neuroinflammation than simply marking activation. To advance this idea, we reassessed data from a model of relapsing-remitting EAE (RR-EAE), which more closely resembles the clinical progression of MS[29]. Thereby, we disclose that *Cd83* expression was elevated during the whole disease course, particularly in remission and recovery phases (Fig. 2g). Furthermore, this approach proved that *Cd83* mRNA is actively translated in microglia during neuroinflammation[30]. Collectively, these data suggest that CD83 expression not only marks early-activated microglial cells

but is also important during the resolution of neuro-inflammatory conditions.

**Expression of CD83 in microglia depends on TGF-β signaling and is differentially regulated by pro- and anti-inflammatory stimuli**
We next investigated how different stimuli affect microglial *Cd83* expression. To this end, we established an in vitro cultivation protocol since microglia tend to change their gene expression profile upon removal from their native environment[5]. Microglia that were treated for 7 days with M-CSF and TGF-β acquired a distinctive phenotype with branched membrane protrusions, which dynamically interact with the environment (Supplementary Fig. 2a, Supplementary Movie 1 and 2). These cultured cells express the microglial marker P2RY12 and also exhibit CD83 promoter activity, highlighted by eGFP-reporter expression (Supplementary Fig. 2b). As microglia depend on intact TGF-β signaling both in vivo and in vitro[5,7,31], we assessed whether this cytokine affects the phenotype of our cultured cells and the expression of *Cd83*. Microglia cultured in the presence of TGF-β had a substantially increased expression of the homeostatic gene *Tmem119*

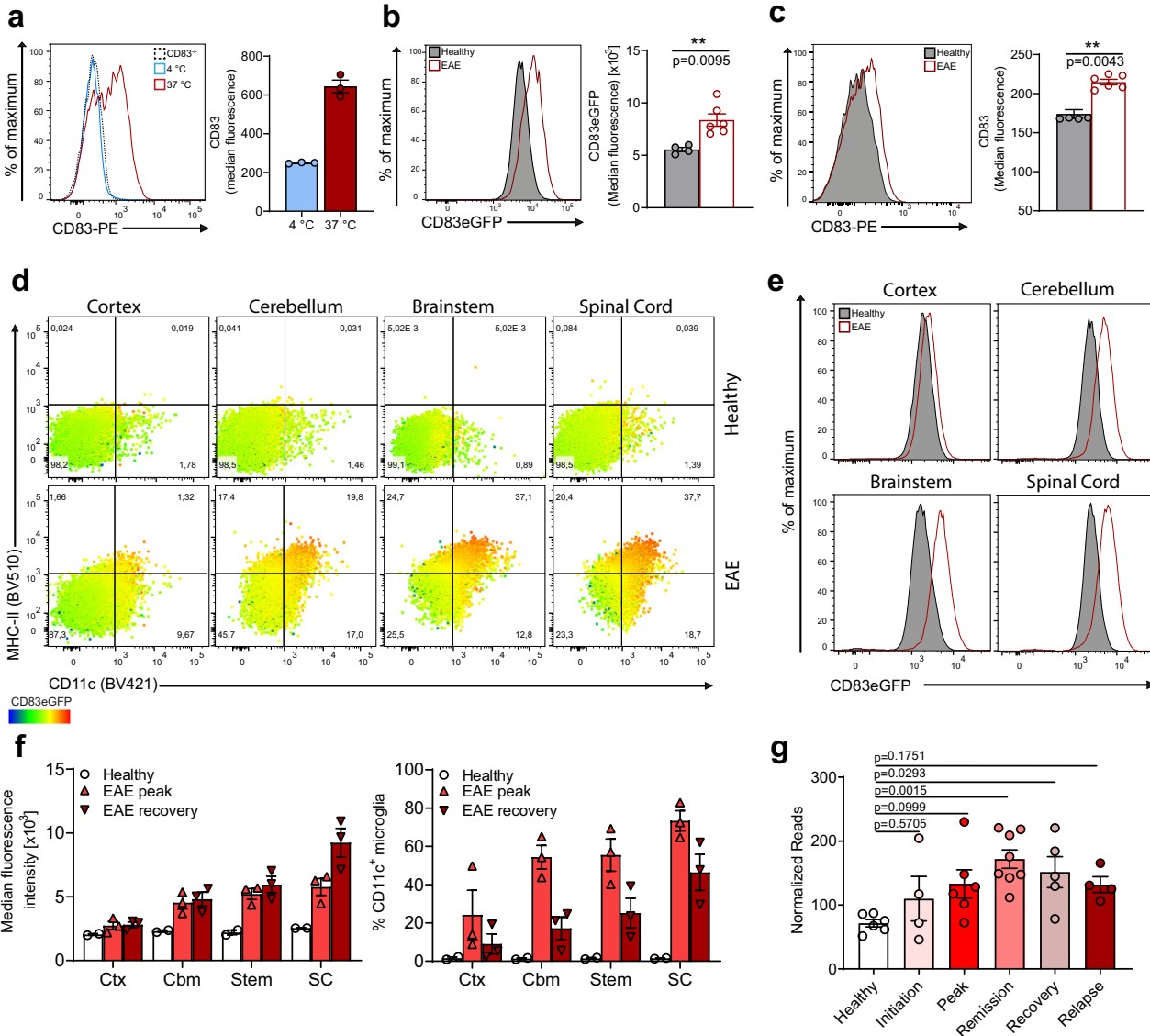

**Fig. 2 | Expression of CD83 in microglia increases with cellular activation but also during resolution of neuroinflammation. a** CD83 expression on microglia upon acute isolation. Freshly isolated microglia were kept either at 4 °C or 37 °C for 6 h (blue and red line, respectively), and expression was assessed via flow cytometry. Cells from CD83[-/-] mice served as negative controls (dotted line) (n = 3 individual mice). **b, c** Analyses of CD83 expression in microglia from healthy (n = 4) or EAE mice (n = 6; peak of disease, day 16 post immunization, p.i.). Statistically significant differences were detected with two-tailed Mann–Whitney U-test. **d** Microglia were isolated from different brain regions of healthy or EAE CD83-eGFP

mice. Cells were assessed for expression of CD11c and MHC-II; dot-plots contain color mapping to visualize eGFP fluorescence and are representative of three individual mice. **e** Representative histogram overlay of eGFP fluorescence shown in (**d**). **f** Comparison between eGFP signal and percentage of CD11c+ microglia in healthy mice, EAE mice at the peak of disease or during the recovery phase (i.e., day 28 p.i.). **g** Longitudinal expression analysis of *Cd83* in RR-EAE mice (n = 4–8 individual mice, sample on the respective disease stage). Statistical significance was calculated with one-way ANOVA with Dunnett's multiple comparison test. In all graphs, data are represented as mean ± SEM.

when compared to cultivation with M-CSF alone, while the activation marker *Msr1* was less expressed (Supplementary Fig. 2c). Interestingly, also *Cd83* expression was considerably increased by the addition of TGF-β, thus resembling a homeostatic, less activated phenotype. To assess the effect of cellular activation, we treated microglial cultures with IFN-γ, TNF-α, or LPS, which are all associated with a pro-inflammatory profile, as well as IL-4, a prototypic cytokine-inducing alternative activation. While IFN-γ had no influence on *Cd83* expression, both TNF-α and LPS significantly decreased *Cd83* mRNA levels (Supplementary Fig. 2d). Interestingly and in sharp contrast, treatment with IL-4 strongly increased *Cd83* expression levels. When we compared the expression pattern of *Cd83* with that of *Tmem119* or *Msr1* in adult microglia cultures, we noticed that pro-inflammatory stimuli

down-modulated *Cd83* and *Tmem119* but had the contrary effect on *Msr1* (Supplementary Fig. 2e). Intriguingly, IL-4, which increases *Cd83* expression, reduces microglial *Tmem119* expression, while enhancing *Msr1*. Thus, regulation of *Cd83* expression in microglia is similar to homeostatic gene transcripts but is also responsive to alternative activation. The opposing effects of TNF-α and IL-4 on *Cd83* expression were also evident in cultures of neonatal microglia, which strongly suggests a context-dependent regulation of *Cd83* expression in murine microglia (Supplementary Fig. 2f). This notion was substantiated further with data from human microglia, which were generated from three different induced pluripotent stem cell lines (iPSCs). Again, we observed a strong induction of *CD83* transcripts upon stimulation with IL-4 but not with TNF-α (Supplementary Fig. 2g).

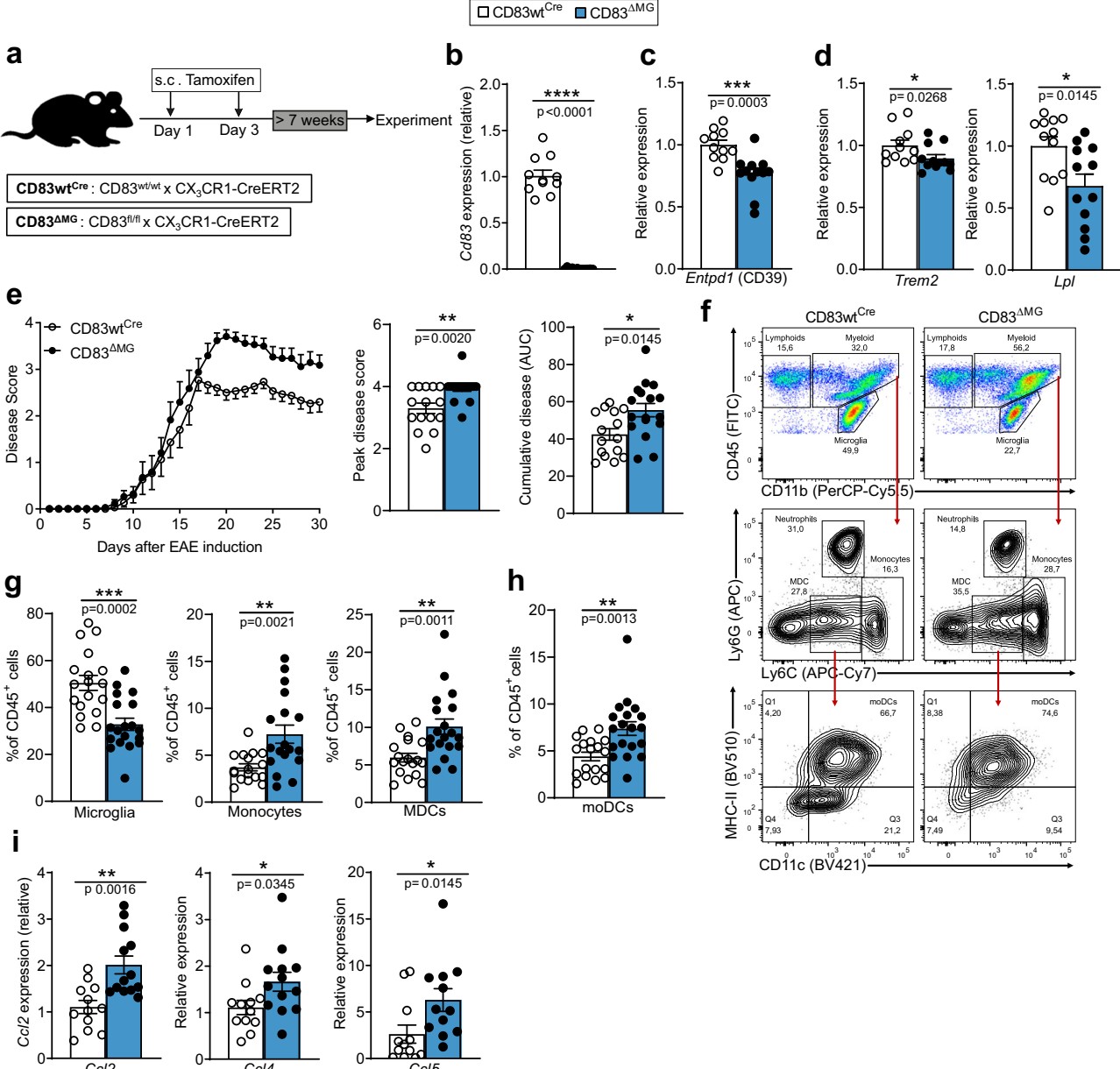

**Fig. 3 | Deletion of CD83 in microglia exacerbates autoimmune neuroinflammation. a** Schematic depiction of conditional knockout strategy.
**b** Verification of successful deletion: microglia were isolated 7 weeks after tamoxifen injection via FACS. After RNA isolation, *Cd83* gene expression was analyzed in CD83^ΔMG cells in relation to controls (*n* = 12, pooled from three independent experiments). **c, d** Gene expression analysis of different gene transcripts in acutely isolated microglia. Expression levels were first normalized to *Rpl4* and relative expression was calculated in relation to controls (*n* = 12, pooled from three independent experiments). **e** EAE course over 30 days, comparison between CD83^ΔMG and CD83wt^Cre controls. Evaluation of the individual peak score of each mouse and the cumulative score, calculated from the area under the curve (AUC) of each

mouse. (*n* = 15 for CD83wt^Cre and *n* = 17 for CD83^ΔMG, data are pooled from three independent experiments). **f** Gating strategy of CNS-infiltrating and resident immune cells (pre-gated on CD45^+ cells) isolated at the peak of disease (day 18 p.i.). **g, h** Relative percentage of microglia, monocytes, monocyte-derived cells (MDCs), and monocyte-derived dendritic cells (moDCs) among all CD45^+ cells at the peak of disease (*n* = 18 for CD83wt^Cre, and *n* = 19 for CD83^ΔMG, data are pooled from three independent experiments). **i** expression of *Ccl2*, *Ccl4*, and *Ccl5* on on day 18 p.i. (*n* = 12 for CD83wt^Cre, and *n* = 13 for CD83^ΔMG, data are pooled from two independent experiments). In all graphs, data are represented as mean ± SEM and two-tailed Mann–Whitney U-test was used to analyze the data.

## Deletion of CD83 in microglia exacerbates autoimmune neuroinflammation

Since our data implied a role for CD83 in microglial activation, we next sought to investigate the impact of CD83 deletion on microglia. To this end, we crossed mice with floxed CD83-alleles to the tamoxifen-inducible CX3CR1-CreERT2 line (hereafter termed CD83^ΔMG). Due to their long-lived nature, microglia are targeted by this approach even after short exposure to tamoxifen (Fig. 3a), and CD83 deletion was evident in microglia 7 weeks after initial tamoxifen treatment (Fig. 3b). It

is well established that peripheral immune cells, such as DCs, are also targeted by CX3CR1-Cre mediated recombination, and we indeed observed a reduction of CD83^+ splenic cDCs by approximately 50% 1 week after tamoxifen injection (Supplementary Fig. 3a). However, 7 weeks after tamoxifen treatment, when all further in vivo experiments commenced, splenic cDC populations have fully recovered CD83 expression, and bone-marrow-derived DCs from these CD83^ΔMG animals also expressed similar levels of CD83 (Supplementary Fig. 3a, b). Thus, we confirmed the microglia-specific action of the inducible CD83cKO

strategy. Since we observed that *Cd83* regulation in microglia shares some characteristics with homeostatic genes, we first investigated the expression of such genes in microglia from CD83^ΔMG mice. We detected a clear but not significant trend toward less expression of *Tmem119* and especially *Siglech* in CD83-deficient microglia (Supplementary Fig. 3c), while transcripts of *Entpd1*, encoding the ectonucleotidase CD39, were significantly reduced (Fig. 3c). Since loss of homeostatic gene expression often coincides with acquisition of a DAM phenotype[32,33], we next investigated the expression of important DAM-genes in CD83-deficent microglia. Surprisingly though, expression levels of DAM-genes *Trem2* and *Lpl* were significantly reduced in CD83-deficient cells while DAM-associated scavenger receptors, such as *Msr1* or *Clec7a*, were unaffected (Fig. 3d, Supplementary Fig. 3d). The same applied for *Apoe*, which cooperates with TREM2 signaling in promoting the DAM phenotype (Supplementary Fig. 3d).

As we have demonstrated, microglial expression of CD83 increases during neuroinflammation and is particularly high at the recovery phase of the EAE model (Fig. 2f, g). In addition, we have recently shown that CD83 expression in DCs restrains autoimmune inflammation in this model[16]. These data prompted the question whether CD83 deletion affects microglial function in the EAE model for neuroinflammation. To approach this issue, we immunized CD83^ΔMG and CD83wt^Cre mice with MOG_{35–55} peptide to induce EAE and monitored disease progression over 30 days. When compared to control animals, CD83^ΔMG mice showed exacerbated and prolonged paralytic symptoms, which is reflected in a significantly higher individual and cumulative disease score (Fig. 3e). To gain insight into the underlying processes, we isolated the CNS (i.e., brain and spinal cord) of EAE mice 18 days after immunization (peak of disease) and analyzed the phenotype and composition of immune cells (CD45^+). Microglia from CD83wt^Cre EAE mice had significantly higher CD83 surface levels than those derived from CD83^ΔMG mice (Supplementary Fig. 4a), thus confirming the KO efficiency. We also verified the KO-specificity in sorted microglia on mRNA level and proved that the KO does not affect infiltrating, peripheral MDCs at the peak of disease (Supplementary Fig. 4b). When assessing the cellular composition of the CNS, we observed a striking infiltration of immune cells of the myeloid lineage (CD45^high/CD11b^+) in CD83^ΔMG mice, which were further stratified into infiltrating neutrophil granulocytes (Ly6G^+/Ly6C^low), monocytes (Ly6G/Ly6C^high), and monocyte-derived cells (MDCs, Ly6G^-/Ly6C^low) (Fig. 3f). While the frequency of microglial cells (CD45^low/CD11b^+) was significantly diminished in CD83^ΔMG mice, the percentage of monocytes and MDCs increased (Fig. 3g). Most of the MDCs expressed high levels of MHC-II and CD11c and were therefore termed monocyte-derived DCs (moDCs), whose proportion among all immune cells was also significantly elevated in CD83^ΔMG mice compared with controls (Fig. 3h). While there was also a significantly increased proportion of cells of the lymphoid lineage (CD45^high/CD11b^−), we observed no differences in the frequency of neutrophil granulocytes (Supplementary Fig. 4c). Since the aggravated EAE phenotype in CD83^ΔMG mice was accompanied by a heightened influx of monocytic cells, we next investigated the production of *Ccl2*, which is a prerequisite for a regular development of EAE symptoms[34]. Compared to control cells, the expression of *Ccl2* was most strikingly elevated in CD83^ΔMG mice (Fig. 3i). We also investigated other chemokines, which are important for EAE development[35], and found that CD83cKO microglia expressed significantly higher levels of *Ccl4* and *Ccl5* at the peak of disease (Fig. 3i). Interestingly, analysis of *Ccl3* expression showed no difference between CD83^ΔMG and controls, demonstrating that chemokine production was not generally elevated due to microglial over-activation (Supplementary Fig. 4d).

### Microglial CD83-deficiency causes an over-activated phenotype supportive of pro-inflammatory pathways

To get a more global impression of the impact of CD83-deletion on microglial function during homeostasis and neuro-inflammatory conditions, we next performed single-cell RNA sequencing (scRNA-Seq) analyses on microglia sorted from the brains of healthy and EAE CD83^ΔMG and control mice, respectively. The workflow of these experiments is shown in Fig. 4a. Briefly, microglia were isolated from both strains, either from naïve mice or at the peak of EAE (day 16 after induction). Then, we employed a droplet-based RNA-Seq approach with four animals per genotype and condition. After initial quality control, integration, and removing clusters of potentially contaminating T cells and macrophages (Supplementary Fig. 5a, b), single-cell transcriptomic profiles for 20,901 cells and 15,984 genes were selected for the analysis. Unsupervised clustering of cells was performed using Seurat's graph-based clustering, and cells were projected on a Uniform Manifold Approximation and Projection (UMAP) to visualize clustering (Fig. 4b). We obtained six clusters, two of which were associated with cells from healthy animals (*Gpr34*^+ and *Ccl4*^+) while the other four cluster largely contained cells from EAE animals (Fig. 4c). Cells in the *Gpr34*^+ subset expressed high levels of *Tmem119*, *P2ry12*, *Siglech*, *and Sall1*, thus representing mainly homeostatic microglia. We also identified two EAE-specific clusters rich in MHC-II related transcripts (*H2-Aa*, *H2-Ab1*, *CD74*), which could be distinguished by high expression of *Apoe* (Fig. 4d). Other clusters that were exclusively present in EAE-derived cells contained genes associated with proliferation (*Mki67*, *Top2a*) or chromatin-assembly (*Hmgb2*, *H2afz*, *Stmn1*). When assessing the distribution of *Cd83* expression, we noticed that while being homogenously present in most identified clusters, *Cd83* was particularly expressed in cells of the *Ccl4*^+ and MHC-II/*Apoe*^+ cluster (Fig. 4e, Supplementary Fig. 5c). Additionally, we confirmed that *Cd83* expression was absent in isolated cells from CD83^ΔMG (Supplementary Fig. 5c).

We further noticed a clear trend that CD83-deficient microglia contributed more to the MHC-II^+/*Apoe*^+ cluster than control cells (Supplementary Fig. 5d). Therefore, we first assessed the activation status of CD83cKO microglia during EAE and detected significantly elevated proportions of activated (i.e., CD11c^+/MHC-II^+) microglia in the CNS of CD83^ΔMG mice (Fig. 5a), paralleling the observed elevated proportion of monocyte-derived APCs in the CNS of these animals. Although MHC-II expression on microglia is regarded as dispensable for the EAE induction or perpetuation[36], the CD11c/MHC-II double-positive phenotype is a typical hallmark of cellular activation during the disease[28]. Thus, CD83-deletion results in amplified activation, a notion that was further corroborated when we assessed the gene expression of microglia sorted from the CNS of EAE mice at the peak of the disease. The exaggerated activation was also reflected by significantly reduced mRNA levels of *Tmem119* in microglia isolated from CD83^ΔMG mice compared to controls (Fig. 5b). Microglia typically lose expression of signature genes such as *Tmem119* during the transition from homeostatic to disease-associated phenotypes[11,37]. Interestingly, CD83-deficient microglia, despite being more activated, exhibited reduced surface levels of CD86 and MHC-II, which is in line with the anticipated role of CD83 in inhibiting MARCH-1-dependent degradation of both molecules (Supplementary Fig. 4e)[15].

To further investigate the effect of CD83-deletion on microglial activation during EAE, we re-analyzed the scRNA-Seq data with the aim to identify possible subclusters within the EAE-derived samples. We calculated the sample-associated relative likelihood with MELD[38] and performed vertex frequency clustering on the merged MHCII+, Stmn1^+, Mki67^+ and Cxcl10^+ clusters, and each major cluster was divided into subclusters based on the visual distribution of the EAE-wildtype associated likelihood. This process resulted in six subclusters, three of which contained cells of both original MHC-II^+ clusters (Fig. 5c). While CD83cKO and WT cells were equally likely distributed in VFC cluster 1, which largely covers the original Cxcl10^+ cluster, WT cells were rather associated with VFC2, and CD83cKO cell with VFC3 (Supplementary Fig. 5e). Interestingly, we detected a striking diversification of WT and cKO microglia when comparing the MHC-II^+ subclusters: WT cells were

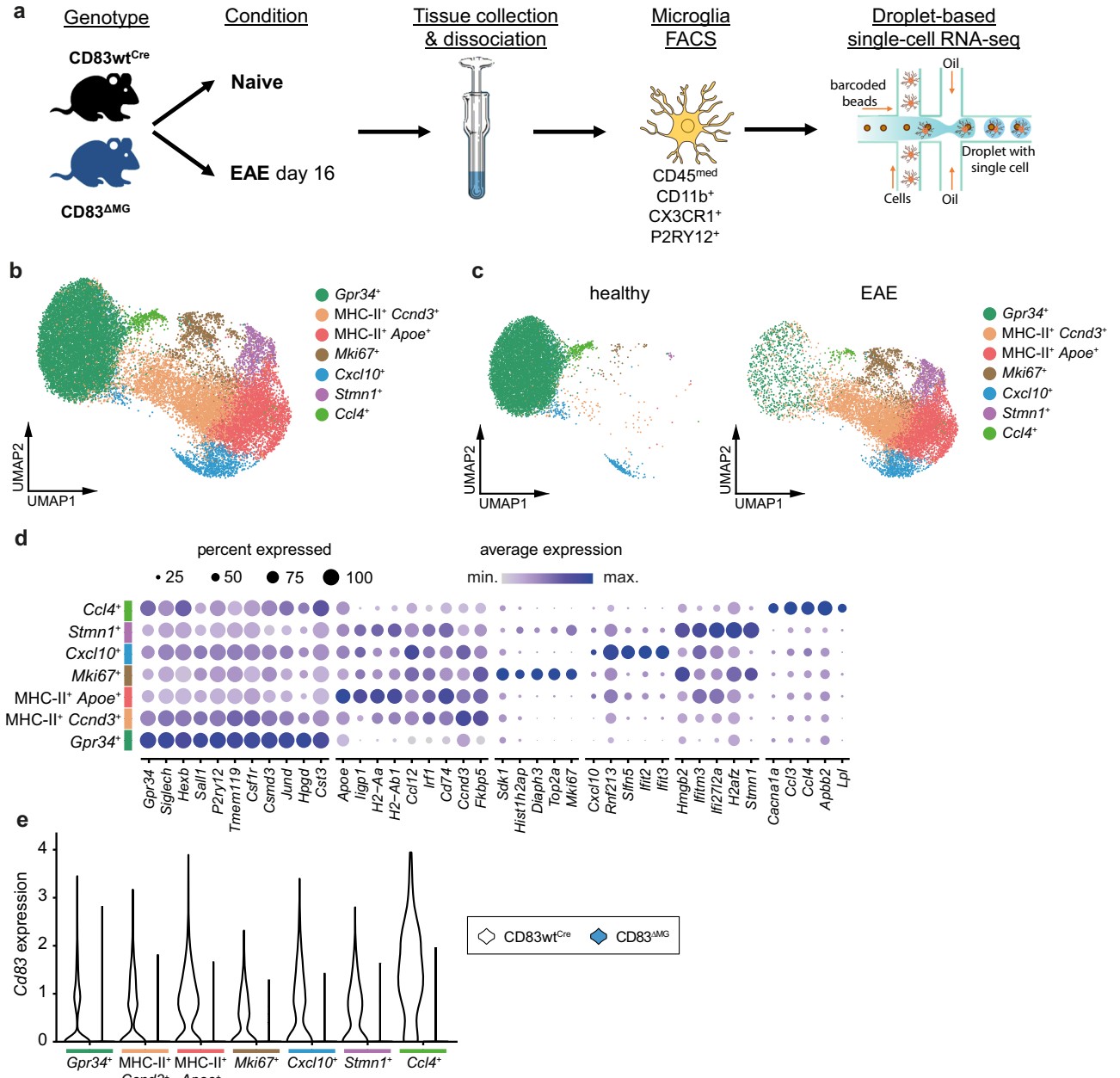

**Fig. 4 | Single-cell RNA-sequencing (scRNA-Seq) of microglia from healthy and EAE mice. a** Experimental setup: naïve and mice at the peak of EAE from CD83wt^Cre and CD83^ΔMG genotypes were used (n = 4 per condition and genotype; total number of animals: 16). Microglia were isolated and subjected to droplet-based scRNA-Seq. **b** UMAP of the filtered and integrated scRNA-seq dataset showing the heterogeneity of microglia in all four conditions: healthy control CD83^wtCre, healthy control CD83^ΔMG, EAE CD83^wtCre, EAE CD83^ΔMG; n = 3 per condition. **c** UMAP shown in (**b**) split between healthy and disease conditions. Clusters *Gpr34*+ and *Ccl4*+ are dominant in healthy animals. The remaining clusters are dominant in diseased animals. **d** Heat-map of the expression of the most relevant marker genes among each microglia cluster from the scRNA-seq dataset shown in (**b**). **e** Violin plots showing *Cd83* expression for each cluster, split between CD83^wtCre and CD83^ΔMG animals.

more likely to be represented in MHC-II^+ sub-cluster 0, whereas CD83-deficient cells had a higher likelihood to be present in MHC-II^+ sub-cluster 2 (Fig. 5d). These subclusters also differed significantly in their gene expression pattern: MHC-II^+ sub-cluster 0 retained more genes of the homeostatic *Gpr34*^+ cluster (e.g., *Siglech, P2ry12, Gpr34*), while MHC-II^+ sub-cluster 2 contained genes like *Apoe, Cd63, Ctsb, Cst7, and Cd74* (Fig. 5e), all of which have been recently assigned to a type of highly activated microglia associated with age-dependent inflammation[39]. Due to the disparity between both MHC-II^+ sub-clusters and the distinct distribution of CD83cKO versus WT cells among them, we reasoned that inflammatory conditions uncover differences between microglia from CD83^ΔMG and WT mice, which were

less pronounced under homeostasis. Therefore, we analyzed microglia from EAE samples for differentially expressed genes (DEGs) between both genotypes as well as associated biological processes and pathways. We discovered that microglia from CD83^ΔMG mice exhibited significantly higher expression of *Apoe, Cst7*, and *Ctsb*, which are associated with the highly activated phenotype (Fig. 5f), whereas control cells rather expressed genes implicated in early inflammatory activation (*Adamts1, Btg1, Egr1, Large1*)[40,41]. When evaluating the Gene Ontology Biological Process terms, we discovered that gene expression pattern in CD83cKO microglia strongly correlates with pathways associated with APC activation and leukocyte-mediated immunity (Fig. 5g). Contrarily, WT cells showed a strong association with

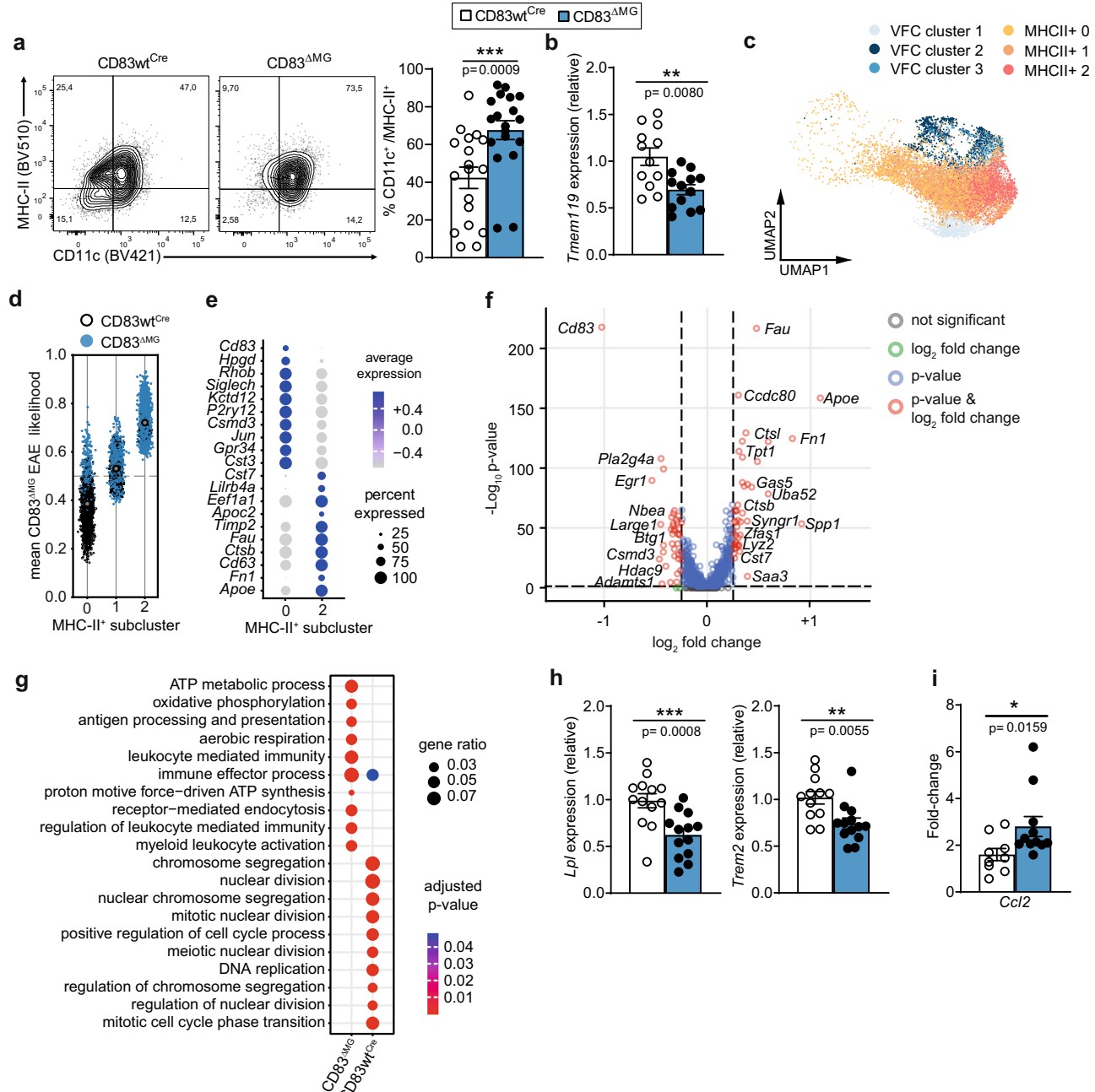

**Fig. 5 | Microglial CD83-deficiency causes an over-activated phenotype supportive of pro-inflammatory pathways. a** Expression of CD11c and MHC-II on microglia from the CNS on day 18 p.i. Representative dot-plots show gating for CD11c⁺/MHC-II⁺ cells, which were quantified in the attached bar chart ($n = 18$ for CD83wt^Cre and $n = 19$ for CD83^ΔMG, data are pooled from three independent experiments). **b** Expression of *Tmem119* on FACS-sorted microglia at the peak of disease (day 18 p.i.; $n = 12$ for CD83wt^Cre and $n = 13$ for CD83^ΔMG, data are pooled from two independent experiments). **c** Clustering of EAE-enriched microglia from CD83^wtCre and CD83^ΔMG animals using Vertex frequency clustering (VFC). Cells of clusters *Gpr34*⁺ and *Ccl4*⁺ (enriched in healthy controls) were excluded from VFC. **d** Mean relative likelihood for each of the MHCII+ VFC clusters to be enriched in the EAE condition in CD83ΔMG animals compared with CD83wtCre animals. **e** Heat-map of the top 10 differentially expressed genes between VFC clusters MHCII⁺ 0 and 2 in EAE. **f** Volcano-plot showing differentially expressed genes between microglia from CD83^wtCre and CD83^ΔMG animals in EAE condition. Log2 fold change threshold was set to 0.25 and adjusted *p*-value threshold to 0.05. **g** Heat-map of the top 10 Gene Ontology Biological Process pathways between CD83wt^Cre and CD83^ΔMG animals in EAE condition based on the differentially expressed genes. **h** Expression of *Lpl* and *Trem2* on microglia on day 18 p.i. ($n = 12$ for CD83wt^Cre and $n = 13$ for CD83^ΔMG, data are pooled from at least three independent experiments). **i** Expression of *Ccl2* in microglial cultures 6 h after treatment with 10 µg/ml myelin debris. Expression is depicted as fold-change over untreated cells ($n = 9$ for CD83wt^Cre and $n = 11$ for CD83^ΔMG, pooled from four independent experiments). In all graphs, data are represented as mean ± SEM and two-tailed Mann–Whitney U-test was used to analyze the data.

pathways involved in cell division. By conducting enrichment analysis of genes whose expression highly correlates with *Cd83* in microglia during EAE, we detected a significant enrichment of processes that are involved in the negative regulation of the immune system and the response to external stimuli (Supplementary Fig. 5f).

Myelin debris derived from destroyed axons can act as a stimulatory agent in the inflamed CNS. Even in the steady state, we have observed that CD83-deficient microglia express less *Trem2* and *Lpl*, which are important for uptake and disposal of myelin debris (see Fig. 3d), and this disparity got even more pronounced in EAE-derived

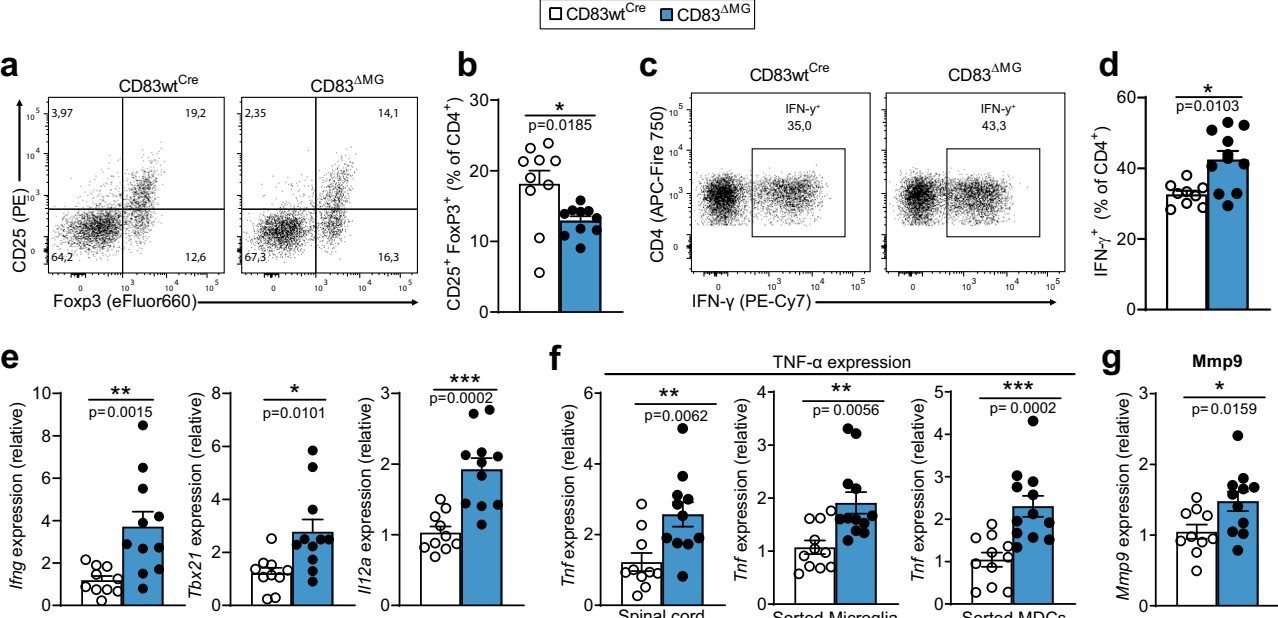

**Fig. 6 | The inflamed CNS of CD83^ΔMG exhibit disturbed T cell balance and a pro-inflammatory, disease-fostering milieu. a, b** Flow cytometric analyses of regulatory T cells (Tregs) in the inflamed CNS. Representative dot-plots (**a**) and quantitative assessment of the percentage of CD25+FoxP3+ among CD4+ T cells (**b**, *n* = 10 for CD83wt^Cre and *n* = 11 for CD83^ΔMG; pooled from two independent experiments) are shown. **c, d** Flow cytometric analyses of IFN-γ producing T cells in the inflamed CNS. Single cells from mice were isolated on day 18 p.i., re-stimulated with PMA/ionomycin for 5 h in the presence of Golgi transport inhibitors and intracellularly stained for IFN-γ. Representative dot-plots (**c**) and quantitative assessment of the percentage of IFN-γ producing CD4+ T cells (**d**, *n* = 10 for CD83wt^Cre and *n* = 11 for CD83^ΔMG; pooled from two independent experiments) are shown. **e** Expression of Th1-related transcripts in the spinal cords of EAE animals on day 18 p.i. RNA was extracted from lumbar parts of spinal cords, reversely transcribed and qPCR was performed for Ifng, Tbx21 (T-bet) and Il12a (*n* = 10 for CD83wt^Cre and *n* = 11 for CD83^ΔMG, pooled from two independent experiments). **f** TNF expression levels in spinal cords and sorted microglia or MDCs derived from EAE mice on day 18 p.i. (*n* = 10 for CD83wt^Cre and *n* = 11 for CD83^ΔMG; pooled from two independent experiments). **g** Mmp9 expression levels in spinal cords of EAE mice on day 18 p.i. (*n* = 10 for CD83wt^Cre and *n* = 11 for CD83^ΔMG; pooled from two independent experiments). Gene expression was normalized to *Rpl4* and relative expression was calculated as ratio between CD83^ΔMG and CD83^wtCre. In all graphs, data are represented as mean ± SEM and Mann–Whitney U-test was used to analyze the data.

microglia (Fig. 5h). Recent studies have shown that Lpl-deficient microglia exhibit an inflammatory profile due to accumulation of intracellular lipid droplets[42]. Thus, we assessed the gene expression of myelin-treated microglia and discovered that CD83-deficient microglia responded with increased expression of *Ccl2* (Fig. 5i). Since Trem2 and Lpl form a functional unit for disposal of accumulating myelin debris in the CNS[43], we examined the ability of CD83-deficient microglia to phagocytose myelin debris and found that microglia from CD83^ΔMG mice were equally capable of ingesting fluorescently labeled myelin (Supplementary Fig. 4f). Collectively, these data provide evidence that neuro-inflammatory events are potentiated by CD83-deficiency through over-activation of microglial cells, most likely due to exaggerated responses to danger signals, such as myelin debris.

**Shifted T cell balance in CD83^ΔMG CNS propagates tissue inflammation**

Above, we have reported that CD83-deficient microglia acquire a highly activated phenotype during EAE and produce chemotactic signals to attract monocytic cells, which ultimately differentiate into APCs. Importantly, the chemokines *Ccl4* and *Ccl5*, which recruit pathogenic T cells[44], were not only significantly elevated in microglia from CD83^ΔMG mice but also in sorted MDCs (Supplementary Fig. 4g). Monocyte-derived APCs are the key cells that interact with encephalitogenic T cells and drive neuroinflammation[11], and thus, we next investigated the T cell subsets infiltrating the CNS of EAE animals. Flow cytometric analyses revealed that the percentage of CD25+/FoxP3+ CD4+ T cells (i.e., regulatory T cells) was significantly diminished in the CNS of CD83^ΔMG mice (Fig. 6a, b). By contrast, the CNS of these animals contained more IFN-γ producing CD4+ T cells, which correlated with

strikingly increased expression levels of Th1-related transcripts (i.e., *Ifng*, *Tbx21*, and *Il12a*) in the spinal cords of CD83^ΔMG mice (Fig. 6c–e). Frequencies of IL-17A and GM-CSF producing T cells were unaltered between both groups (Supplementary Fig. 4h). In addition, the distorted ratio of Tregs and Th1 cells was only overt at the site of inflammation (i.e., the CNS) but not in peripheral lymph nodes (Supplementary Fig. 4i). Excessive amounts of IFN-γ are known to impair neuroprotective features of microglia, in part by upregulation of TNF-α production, which interferes with oligodendrogenesis[45,46]. Thus, we analyzed the expression pattern of *Tnf* in spinal cords of EAE mice as well as microglia and MDCs sorted from the CNS at the peak of the disease. We disclosed drastically elevated *Tnf* transcripts in spinal cords of CD83^ΔMG mice. Both microglia and MDCs from CD83^ΔMG mice expressed significantly higher *Tnf levels* than cells from control animals (Fig. 6f). TNF-α not only directly affects the oligodendrocyte-dependent remyelination process but also promotes the destruction of the blood-brain-barrier by inducing the expression of matrix metalloproteinase 9 (MMP9), for instance in astrocytes[47]. Thus, we assessed the spinal cords of EAE mice for the expression of *Mmp9* and detected significantly elevated transcript levels in CD83^ΔMG mice, demonstrating that CD83-deletion in microglia leads to more severe disruptive processes in the inflamed CNS (Fig. 6g). Other pro-inflammatory transcripts like *Il6* and *Il23a* were not differently expressed in the spinal cord between both groups (Supplementary Fig. 4j), proving that the elevated expression of TNF-α and MMP9 is not a result of generally augmented inflammation in the CD83^ΔMG CNS.

In summary, we have provided experimental evidence that microglial CD83 expression modulates the reaction to neuro-inflammatory damage in the CNS. CD83-deficient microglia mount

an excessive pro-inflammatory response, marked by elevated production of chemokines, which recruit pathogenic cells to the site of inflammation. Together, these cells create a toxic environment fostering tissue damage and thus perpetuating paralytic symptoms. These data disclose that CD83 expression by microglia is not simply a marker molecule but rather a gatekeeper of cellular activation during neuroinflammation.

## Discussion

Microglia are undisputed central players in neuronal immune processes. They fulfill a plethora of tasks during CNS homeostasis but also critically affect the course of neurologic disorders like neurodegeneration and neuroinflammation[48]. In this study, we disclosed that the CD83 molecule acts as a regulatory node governing microglial activation during neuroinflammation. We also provide evidence that physiological *Cd83* expression in microglia is subject to regional heterogeneity with the highest expression levels in the spinal cord, which is in line with previously published single-cell data[27]. Interestingly, those regions with higher *Cd83* expression consist mainly of white matter. In human microglia, *CD83* expression is particularly high in cells from white matter versus gray matter[23], a finding which we confirmed on the protein level (see Fig. 1e). Interestingly, we observed an association of CD83 with a rather ameboid cell shape, indicative of activation. Human white matter microglia show elevated activity of the NF-κB pathway, which subsequently could activate *CD83* transcription via specific binding sites present in the CD83 promoter[23,49].

We also demonstrated that the regional expression pattern of Cd83 is retained under inflammatory conditions. Using a murine model for neuroinflammation, the EAE, we revealed that microglial expression of *Cd83* increases especially in regions associated with white matter. While cortical regions are not severely affected, EAE-induced demyelination mainly pertains cerebellum, brainstem, and spinal cord[50], where *Cd83* expression strongly increases during EAE. Additionally, augmented *Cd83* promoter activity coincides with cellular activation (i.e., CD11c+/MHC-II+ cells). This confirms recent data showing that during homeostasis, *Cd83* is expressed in "pre-activated" or immune-alert microglial cells and that its expression increases in early phases of neuroinflammation[11,26,27]. In this respect, it is intriguing that *Cd83* expression even proceeds when the cells re-acquire a less activated state, especially in spinal cord microglia. Spinal cord microglia have been demonstrated to exhibit a different gene expression pattern than cortical microglia, but these changes are mostly unaffected during EAE[27,51]. The prolonged expression of *Cd83* in spinal cord microglia suggests that *Cd83* is not only associated with acute activation but also with a reparative phenotype during the resolution phase of EAE. In line with this, *Cd83* expression is detected in microglia after cuprizone-induced demyelination and remains elevated during remyelination[52].

Interestingly, microglia react with changes in *Cd83* expression after stimulation in a unique manner: unlike other immune cells, microglia do not respond with elevated *Cd83* expression after activation with pro-inflammatory mediators like LPS or TNF-α[15]. This regulation seems contradictive to the observed elevation of CD83 during EAE, but we provided evidence that CD83 expression during EAE is not only associated with activated microglia but rather even increases during the resolution phase. In line with this notion, alternative activation with IL-4 strongly induces *Cd83* transcription in microglia, which has been also observed in bone-marrow-derived macrophages[53]. Interestingly, IL-4 acts differently on the microglial expression of *Tmem119* and *Cd83*. Lentiviral CNS delivery of IL-4 induces microglial proliferation and a phenotype reminiscent of embryonic microglia[54]. Such proliferative region-associated microglia were shown to express less *Tmem119* and more *Cd83* (ref. 14; accessed via https://myeloidsc. appspot.com/), mirroring the results obtained with IL-4 treatment of our in vitro cultures. In addition, alternatively activated microglia serve

to induce oligodendrocyte differentiation during remyelination[46,55], suggesting a pro-resolving function of CD83+ microglia. This connection is further substantiated by the fact that we observed less *Lpl* expression in CD83-deficient microglia from either healthy or EAE mice since *Lpl* expression is also associated with an alternatively activated phenotype[56]. Additionally, Lpl-deficient microglia have been shown to accumulate excessive amounts of lipid droplets, which deteriorates microglial function and induces a pro-inflammatory transcriptional program[42,57]. We also detected less *Trem2* expression in CD83-deficient microglia both in the steady state and under inflammatory conditions. Trem2-KO mice show a defective response to myelin debris in the cuprizone demyelination model and are incapable of inducing *Lpl* expression, amongst other transcripts involved in lipid metabolism[58]. Interestingly, another study on Trem2-KO mice has reported that microglia from these mice express less *Cd83*[59]. This implies an important regulatory circuit between CD83, Trem2, and Lpl in microglia, which are challenged with myelin debris, for instance, after neuroinflammation-induced demyelination. By performing scRNA-Seq analyses, we discovered a microglial cell cluster that expressed particularly high levels of *Cd83* (Ccl4+ cluster, see Fig. 4d, e). This cluster not only contains cells with high *Lpl* expression, but its gene expression pattern also resembles a specific cluster in human microglia (high expression of *CCL3* and *CCL4*)[22]. Therefore, these data suggest a similar and evolutionary conserved regulation of CD83 expression in microglial cells.

Microglia undergo a series of phenotypic and functional changes during neuroinflammation[11]. Elevated expression of *Cd83* was implied in the early activation of microglia, which was confirmed within this study. However, *Cd83* expression not only coincides with early activation but is also elevated during chronic and remission disease phases. Other studies provided hints that CD83 is linked to cuprizone-induced demyelination[52], 'pre-activated' human microglia[12,22], and microglia from human white matter[23]. These data also suggest that CD83+ microglia play an important role in demyelinating diseases. Here, indeed, we show that prolonged CD83 expression is associated with a pro-resolving state of microglia. CD83-deficient microglia lapse into an over-active state, in which they attract inflammatory monocytes from the circulation and create a disease-promoting inflammatory environment. While CD83cKO cells are equally capable of phagocytosis, they react with altered expression of chemokines and gene transcripts involved in the degradation of lipids. Consequently, CD83 might not only mark activated cells but also be indicative of cells important for the clearance of debris and initiating the resolution of inflammation. Similarly, we have described that CD83-deficiency in DCs and macrophages subverts their capacity to regulate immune responses, eventually resulting in aggravated autoimmunity and hampered resolution of inflammation[16,18]. In this respect, it is worth mentioning that CX3CR1-CreERT2-driven depletion of *Cd83* can affect border-associated macrophages, which were also shown to express *Cd83*[20]. Although we have demonstrated that disease severity in CD83^ΔMG mice clearly correlates with an exaggerated activation status of CD83-deficient microglia, we cannot fully exclude a contribution of targeted border-associated macrophages to the observed phenotype.

The exaggerated neuroinflammation in CD83^ΔMG mice is also characterized by a shifted ratio between FoxP3+ Tregs and IFN-γ producing Th cells in the inflamed CNS. Interestingly, high doses of IFN-γ subvert the capacity of microglia to induce Tregs in vitro[60]. This effect is also sensitive to the level of MHC-II on the microglial surface, as higher MHC-II expression inversely correlates with the proportion of induced antigen-specific Tregs. We also observed an increased proportion of CD11c+/MHC-II+ microglia in CD83^ΔMG mice, which, together with heightened Th1-responses, could account for the observed reduction of Tregs and worsened disease outcome. Although the role of IFN-γ for the pathogenesis of EAE is still disputed[61], Th1 cells are equally capable of inducing paralytic symptoms upon adoptive

transfer compared to Th17 cells[62]. Furthermore, a unique IFN-γ producing T cell population is present in spinal fluids of MS patients, which is specifically enriched for the expression of MMP9[63]. This subset expresses the chemokine receptors CCR2 and CCR5, which respond to CCL2 and CCL4/5, respectively. CD83-deficient microglia express significantly more CCL2 and CCL5 during EAE, and we detected increased levels of Th1-related transcripts as well as *Mmp9* expression in the spinal cords of CD83ΔMG mice. This provides an interesting link to human data and suggests that over-activated CD83-deficient microglia recruit distinctive pathogenic T cells, which exacerbate the disease course.

Collectively, these data identify CD83 as an important regulator of microglial function under neuro-inflammatory conditions. Due to their incapability to properly remove myelin debris, CD83-deficient microglia react with an overshooting production of chemokines, which subsequently attract peripheral immune cells to the inflamed CNS. The interplay with these cells then creates an even more destructive environment, fostering the inflammation and impeding proper resolution of inflammation (Fig. 7). Thus, microglial CD83 expression is not only a marker of cellular activation but an important key regulatory molecule, which keeps neuro-inflammatory processes in check.

## Methods
### Animals
CD83eGFP reporter animals were generated and kept in-house[24]. CD83fl/fl mice were generated as described previously[16] and were bred to *Cx3cr1CreErt2/+(Jung)* mice for microglia-specific depletion of the *Cd83* gene (CD83ΔMG). *Cx3cr1*-CreERT2 mice were purchased from Jackson Laboratories. CD83wt mice, which were bred with *Cx3cr1*-CreERT2

mice, served as control animals in all experiments (hereafter termed as CD83wtCre). For in vitro assessment of CD83-deficient microglia, we used conditional knockout mice generated with the *Cx3cr1*-Cre line and the appropriate Cre-mice as controls. We used an equal proportion of age-matched male and female mice in all experiments. Mice were kept under specific pathogen-free conditions at 22 °C and 40–50% humidity on a 12/12-h light/dark cycle and were fed with standard laboratory chow ad libitum. Animal care and all experimental procedures of the present study were performed in accordance with the European Community Standards on the Care and Use of Laboratory Animals and were approved by the local ethics committee (Administration of Lower Franconia; Reference number 55.2.2-2532-2-1193).

### Tamoxifen treatment
To induce a microglia-specific deletion of the *Cd83* gene, CD83ΔMG and control mice received a subcutaneous injection of 4 mg of tamoxifen (Sigma #T5648), which was solved in warm corn oil (Sigma #C8267) at a concentration of 20 mg/ml, and this procedure was repeated 48 h later. To ensure the reconstitution of CD83 expression in peripheral CX3CR1+ cells, all experiments were conducted at least 7 weeks after the last tamoxifen treatment.

### EAE induction
Mice were immunized subcutaneously with 100 μl of emulsified complete Freund's adjuvant (CFA, Sigma) supplemented with 10 μg/μl *Mycobacterium tuberculosis* and 100 μg MOG35–55 peptide (Charité Berlin) and received intraperitoneal injections of 200 ng pertussis toxin (List Labs) at the time of immunization and 48 h later. Mice were

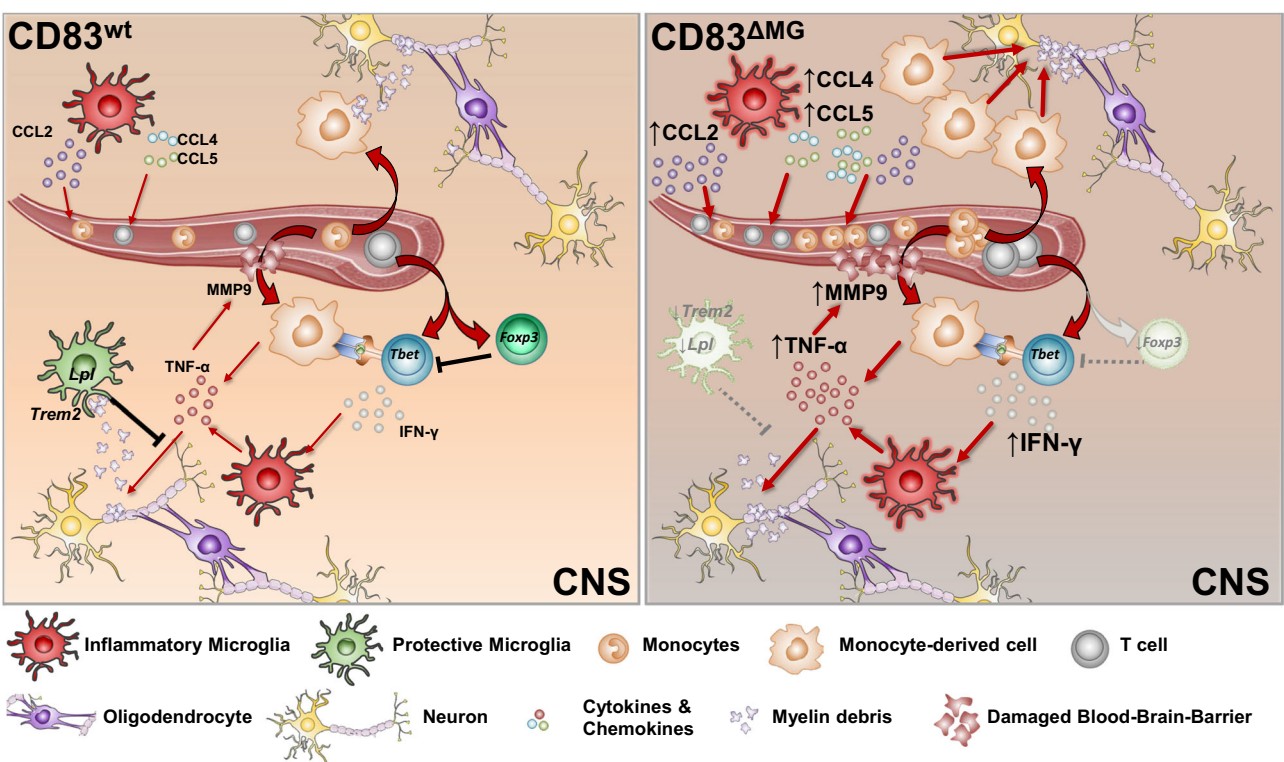

**Fig. 7 | Graphic summary.** During neuroinflammation, CD83ΔMG microglia get over-activated due to incapability to cope with myelin debris, which leads to enhanced production of chemokines (e.g., CCL2). Consequently, the CNS of CD83ΔMG mice experiences an elevated influx of pathogenic monocytes and T cells, the latter of which rather differentiate to Th1 cells than to regulatory T cells. The elevated amount of IFN-γ induces TNF-α production from microglia as well as monocyte-derived cells, which causes elevated expression of MMP9 and subsequent damage to the blood-brain barrier. On the other hand, CD83-deficient microglia fail to upregulate sufficient amounts of *Lpl* and *Trem2* to clear accumulating myelin debris, thus perpetuating the pro-inflammatory milieu instead of leading to the resolution of inflammation.

monitored every other day for signs of clinical manifestation of EAE, and clinical scores were attributed as follows: 0, no disease; 1, tail weakness; 2, hind limb weakness; 3, hind limb paralysis; 4, hind limb paralysis with forelimb weakness; 5, moribund or dead animals.

## Isolation of primary microglia and brain-infiltrating cells for flow cytometry

Microglia and other immune cells within the CNS were isolated by a one-step density centrifugation protocol as described previously, with some modifications[64]. Briefly, mice were euthanized with $CO_2$ inhalation and subsequently perfused with ice-cold PBS by cardiac puncture, and brains were collected and placed in RPMI medium (Thermo Fisher Scientific) on ice. Tissue was homogenized with a tissue dounce grinder set. Homogenates were resuspended 37% Percoll™ (Sigma) and subjected to density centrifugation ($550 \times g$, 18 min) to remove myelin debris. Cells were collected in 1x HBSS (Thermo Fisher Scientific) and used for flow cytometry. Microglial cells were defined as $CD11b^+CD45^{int}CX3CR1^+$, whereas the $CD45^{hi}$ population represented infiltrating lymphoid ($CD11b^-$) and myeloid ($CD11b^+$) cells.

## Primary microglia culture

Perfused brains were minced with scalpel blades and transferred to C-tubes (Miltenyi Biotech) containing an enzymatic digestion mix consisting of DMEM/F12 (Thermo Fisher Scientific) supplemented with 1 mg/ml papain (#P, 2 U/ml Dispase II, and 20 U/ml DNase I (all from Sigma). Tissue was homogenized at 37 °C using a gentleMACS™ Octo Dissociator with Heaters (Miltenyi Biotech). Myelin debris was removed by density centrifugation of the homogenate over a discontinuous 70%-37%-30% Percoll gradient. Microglial cells were collected from the 70/37-interphase, resuspended in Microglia Culture Medium (MCM, consisting of DMEM/F12+1% Pen/Strep/Glutamine, 10% FCS) and seeded onto 12-well plates coated with 0.1 mg/ml Poly-D-Lysine (Merck). After incubation for 60 min at 37 °C, non-adherent cells were removed and the medium was exchanged with MCM, supplemented with 10 ng/ml recombinant M-CSF and 2 ng/ml TGF-β (BioLegend). The medium was renewed every other day, and cells were stimulated at day 6 of culture. Stimulation included 300 U/ml IFN-γ, 500 U/ml TNF-α, 40 ng/ml IL-4 (all Peprotech), 10 ng/ml LPS (Sigma), or 10 μg/ml myelin debris (for preparation, see below).

## Neonatal microglia culture

Mice aged P0–P3 were euthanized by decapitation, and brains were dissected into PBS on ice. Brains of 6–8 mice were pooled, centrifuged at $500 \times g$ for 10 min at 4 °C and resuspended in 0.25% Trypsin-EDTA (Thermo Fisher Scientific) at 37 °C for 10 min. DNase I (Thermo Fisher Scientific) was added at 1 mg/ml to the solution, and the brains were digested for 10 more minutes at 37 °C. Trypsin was neutralized by adding DMEM + GlutaMAX (Thermo Fisher Scientific) supplemented with 10% FBS (Thermo Fisher Scientific) and 1% penicillin/streptomycin (Thermo Fisher Scientific), and cells were passed through a 70 μm cell strainer. Cells were centrifuged at $500 \times g$ for 10 min at 4 C, resuspended in DMEM + GlutaMAX with 10% FBS 1% penicillin/streptomycin and cultured in T-75 flasks (Sarstedt), pre-coated with 2 μg/ml Poly-L Lysine (PLL, Provitro) at 37 °C in a humidified incubator with 5% $CO_2$ for 5–7 days until confluence was reached. Mixed glial cells were shaken for 30 min at 180 rpm, the supernatant was collected and the medium was changed, and then cells were shaken for at least 2 h at 220 rpm and the supernatant was collected and the medium was changed again. CD11b+ microglia were isolated from the collected supernatant using the CD11b Microbead Isolation kit (Miltenyi) according to the manufacturer's instruction and seeded onto PLL-coated plates.

## Human iPSC-derived microglia culture

Three different human iPSC lines (KOLF2.1, from The Jackson Laboratory; BIONi010-C and BIONi037-A, from the European Bank for Induced Pluripotent Stem Cells) were used to generate microglia-like cells using a well-established protocol[65]. Briefly, $2.5 \times 10^6$ iPSCs were seeded into an AggreWell™800 24-well plate (STEMCELL Technologies) to allow embryoid body (EBs) formation and were fed daily with mTeSR+ medium (STEMCELL Technologies) supplemented with 50 ng/ml BMP4 (Miltenyi Biotec), 50 ng/ml VEGF (Miltenyi Biotec), and 20 ng/ml SCF (R&D Systems). After 4 days, the EBs were differentiated in 6-well plates in X-VIVO15 medium (Lonza) supplemented with 100 ng/ml M-CSF (Miltenyi Biotec), 25 ng/ml IL-3 (Miltenyi Biotec), 2 mM Glutamax (Invitrogen Life Technologies), and 0.055 mM β-mercaptoethanol (Thermo Fisher Scientific), with weekly medium replenishment. Microglial precursors emerging in the cell culture supernatant after approximately 1 month were collected and seeded at a concentration 50,000/$cm^2$ for further differentiation into microglia-like cells in Advanced DMEM/F12:Neurobasal medium (both Gibco) supplemented with 1x B27 supplement plus Vitamin A (Thermo Fisher Scientific), 2 mM GlutaMAX, 0.055 μM β-mercaptoethanol, 20 ng/ml M-CSF and 100 ng/ml IL-34 (PeproTech) for 14 days with medium change twice a week. The cells were stimulated on day 13 of culture with either 500 U/ml TNF-α or 40 ng/ml IL-4 (Peprotech).

## Myelin isolation and staining

Myelin was isolated as previously described[66]. In brief, myelin was isolated after mechanical homogenization of perfused brains and washed twice with PBS containing 0.32 M sucrose. Afterward, the CNS homogenate was carefully transferred on top of a 0.85 M sucrose solution in PBS. The CNS sucrose gradient was centrifuged at $3200 \times g$ for 60 min. The interphase containing myelin was isolated and subjected to osmotic shock by washing it twice in distilled water. The protein amount of purified myelin was assessed using a Pierce™ BCA Protein Assay Kit, and 1 mg of myelin was stained with pHrodo dye (all from Thermo Fischer Scientific) in PBS pH 8.2, according to the manufacturer's instructions, followed by washing in PBS pH 7.4 to remove excess dye.

## Flow cytometry analysis and cell sorting

Surface staining was performed in PBS containing LIVE/DEAD™ Fixable Aqua Dead Cell Stain (Thermo Fisher Scientific) for 30 min at 4 °C. For intracellular staining, cells were fixed and permeabilized, and intracellular staining was performed in Permeabilization Reagent (Thermo Fisher Scientific). Antibodies (all from BioLegend, unless otherwise stated) against following antigens were used with the indicated dilutions: CD3 (17A2; 1:200), CD4 (RM4-5; 1:200), CD8 (53-6.7; 1:100) CD11b (M1/70; 1:100), CD11c (N418; 1:100), CD25 (PC61; 1:200), CD45 (I3/2.3; 1:100), CD83 (Michel19; 1:50), CD86 (GL-1; 1:200), CX₃CR1 (SA011F11; 1:200), FoxP3 (FJK-16s, *eBioscience*; 1:100), IFN-γ (XMG1.2; 1:200), IL-17A (TC11-18H10.1; 1:300), I-A/I-E (M5/114.152; 1:200), Ly6C (HK1.4; 1:400), Ly6G (1A8; 1:100), P2RY12 (S16007D; 1:100). Expression was assessed with a FACSCantoII flow cytometer (BD) and analyzed with FlowJo software version 10 (FlowJo, LLC). Cells were pre-gated on $CD45^+$ for further analyses.

## Generation and sequencing of droplet-based single-cell RNA sequencing libraries

Microglia single-cell suspensions were generated from the brain of healthy or EAE mice (day 16 after induction) as described above. Surface staining was performed in PBS containing BioLegend CD45 (I3/2.3), CD11b (M1/70), CX3CR1 (SA011F11) and P2RY12 (S16007D) antibodies for 30 min at 4 °C. The antibody cocktail was supplemented with hashtag antibodies (TotalSeq-B0301/B0302/B0303/B0304 anti-mouse hashtag 1/2/3/4, Biolegend) to label individual animals. Microglia were isolated by fluorescence-activated cell sorting (FACS) for the presence of CD45 (intermediate), CD11b, CX3CR1 and P2RY12 signals. Cells were sorted from individual animals, washed, counted and concentrated to 1000 cells/μl before pooling. Four independent pools

were generated, each containing equal numbers of cells from healthy CD83wt[Cre], healthy CD83[ΔMG], EAE CD83wt[Cre] and EAE CD83[ΔMG] animals (16 animals in total, 4 biological replicates per condition in 4 pools, 1 pool containing all 4 conditions). Up to 25,000 cells (hyper-loading) pooled from four animals were loaded into a single well of a Chromium chip G (10x Genomics). 3' gene expression libraries were generated using Chromium Next GEM Single Cell 3' Kit 3.1 with 3' Feature Barcode Kit and dual indexing (10x Genomics protocol CG000316 Rev D). Libraries were sequenced as PE150 by Illumina sequencing to approximately 65% saturation. Reads were mapped to the mm10 mouse genome (GENCODE vM23/Ensembl 98) using the 10x Genomics Cell Ranger pipeline (6.0.0) with default settings.

### Quality control, normalization and integration of single-cell RNA sequencing libraries

Single-cell RNA sequencing datasets of murine microglia from four sequencing libraries went through initial quality control to exclude low-quality cells by examining the distribution of the number of unique molecular identifiers (UMI) (>1500), the number of genes detected (>1000), the mitochondrial ratio (<0.04), and the overall complexity (log10(genes/UMI) > 0.8). Using Seurat (4.1.1)[67,68] and R (4.2.1), datasets were subjected to hashtag antibody oligos (HTO) separation using Seurat's "HTODemux" with a 0.95 threshold to separate conditions and genotypes and to remove doublets and negative cells. In this step, the second sequencing library (pool 2) was discarded as the expression of the antibody was not enough to separate conditions. A second quality control was performed, and cells with UMI < 4000 and number of expressed genes <2000 were excluded. Genes expressed in less than ten cells were discarded. To remove the batch effect, Harmony[69] integration was performed on Seurat's log-normalized and scaled data while regressing out cell cycle scores (Supplementary Fig. 5a). In this step, a set of genes that lacked biological relevance to our research and displayed uneven expression patterns (Supplementary Table 1), thereby exerting an impact on the clustering results, were removed from the dataset. Seurat's graph-based clustering and UMAP visualization based on the first 30 Harmony components resulted in identifying contaminating clusters of T cells and macrophages (Supplementary Fig. 5b). These clusters were discarded, and the workflow of integration and clustering (resolution = 0.5) was repeated. To perform differential gene expression, each dataset was processed individually using the "SCTransform" (v2) workflow[70] and "PrepSCTFindMarkers" was applied to the SCT assays to eliminate the impact of fluctuating sequencing depths. To identify the markers of each cluster, Seurat's "FindConservedMarker" was used, and genes with adjusted $p$-value < 0.05 in all conditions were considered significant. The overall log fold change (LFC) was calculated by averaging the LFC over conditions for each gene.

### Differential abundance analysis

To conduct the cluster differential abundance test, edgeR (3.38.1)[71] was used. The input for edgeR consisted of cluster counts across all conditions and replicates, which were subsequently subjected to "estimateDisp", "glmQLFit", and "glmQLFTest"[72]. Significance was determined based on a false discovery rate threshold of <0.05 for each test.

### Sample-associated relative likelihood (MELD)

To evaluate the impact of $Cd83$ knockout on cellular composition during EAE, we employed MELD (version 1.0.0, implemented in Python 3.8.8)[38]. PHATE dimension reduction was performed, followed by PCA. By running "meld" with default settings, we computed densities associated with replicates. The resulting densities were normalized (using L1 normalization to ensure that the sum of each cell's values equals 1) and averaged across all replicates for each condition per cell. Condition-associated relative likelihood in each graph-based cluster

was visualized. Next, vertex frequency clustering (VFC) was performed on merged MHCII+, proliferating and $Cxcl10$+ clusters, and each major cluster was divided into subclusters based on the visual distribution of the EAE-wildtype associated likelihood.

### Functional enrichment analysis

Functional enrichment analysis was performed using Clusterprofiler (4.4.4)[73] "compareCluster" and "enrichGo" and by setting the ontology to Gene Ontology Biological Processes (GOBP) and the "universe" to all genes detected in the dataset.

### Gene-gene correlation analysis

To robustly calculate the gene-gene correlations, hdWGCNA (0.2.02)[74] was used by grouping transcriptionally similar cells in each replicate and sub-cluster of Microglia from EAE CD83wt[Cre] condition. To remove noise, "TestSoftPower" was used to determine the value to which the correlations were increased. The resulting correlation matrix was used to extract the correlation values of all genes with $Cd83$ and the top ten percent of those genes with higher correlation values were selected as highly correlating with $Cd83$ and subjected to functional enrichment analysis.

### qPCR

Cells were lysed with RLTPlus buffer and stored at −80 °C until RNA isolation. Total RNA was isolated using the RNeasy Plus Micro Kit (Qiagen), and mRNA was transcribed into cDNA using First Strand cDNA Synthesis Kit (Thermo Fisher Scientific). Gene expression was analyzed via qPCR and the Blue S'Green qPCR Kit (Biozym) and normalized to the expression of $Rpl4$. Raw data were analyzed with the BioRad CFX Maestro V 2.3 software. For human iPSC-derived microglia, we lysed the cells in LB1 buffer and isolated RNA using the NucleoSpin Kit (Macherey Nagel). Afterward, mRNA was transcribed into cDNA using High-Capacity cDNA Reverse Transcription Kit (Applied Biosystems). Gene expression was analyzed via qPCR and the SYBR Master Mix (Applied Biosystems) and normalized to the expression of $RPLO$. To compensate for inter-assay variability, relative expression levels were calculated as a ratio of CD83[ΔMG] to CD83wt[Cre], or as fold change in relation to unstimulated cells. All primers were designed and validated according to the MIQE guidelines (see Table 1).

### Immunofluorescence microscopy

On day 6 after isolation, cell cultures of primary microglial cells were washed with warm PBS. Cells were fixed with 4% paraformaldehyde (PFA) in PBS for 10 min at 37 °C and subsequently blocked for 30 min with PBS containing 5% mouse serum. Cells were immunostained with P2RY12 (clone S16007D, Biolegend, 1:100) overnight at 4 °C, followed by staining with secondary goat-anti-rat-AlexaFluor594 antibody (clone Poly4054, Biolegend, 1:250), for 60 min. Staining of nuclei was performed with a DAPI-containing mounting medium (Roti®Mount-FluorCare, Carl Roth). Immunofluorescence images were acquired on a Keyence BZ-X800 fluorescence microscope.

Human neocortical brain tissue sections were prepared from access cortical tissue outside the tumor area, resected in order to gain access to the pathology, obtained from a patient undergoing tumor surgery (age at surgery: 60, sex: F, resected brain area: frontal lobe, diagnosis: WHO Grade 4). Approval (# 147/2021BO2) of the ethics committee of the University of Tübingen as well as written informed consent was obtained from the patient, allowing spare tissue from resective surgery to be included in the study. The brain tissue block was fixed in 4% PFA overnight at 4 °C and then washed thoroughly with PBS to remove excess PFA. Fixed tissue was cut into 25-μm-thick sections using a freezing-sliding microtome (Slee Technik GmbH). Sections were collected in a 12-well plate and stored in cryoprotectant solution (35% ethylene glycol and 25% glycerol in PBS) until further processing. Following antigen retrieval (Citrate Buffer, 20 min, 90 °C), tissue sections were blocked for 2 h at RT

**Table 1 | Primers used for qPCR**

| Gene | Orientation | Sequence |
|---|---|---|
| **Apoe** | Forward | 5′-TCAGCTCGAGTGGCAAAG-3′ |
| | Reverse | 5′-TTACTTCCGTCATAGTGTCCTC-3′ |
| **Ccl2** | Forward | 5′-GAGAGCTACAAGAGGATCACC-3′ |
| | Reverse | 5′-GATCTCATTTGGTTCCGATCC-3′ |
| **Ccl3** | Forward | 5′-TCTGTCACCTGCTCAACATC-3′ |
| | Reverse | 5′-CGATGAATTGGCGTGGAATC-3′ |
| **Ccl4** | Forward | 5′-GCTGTTTCTCTTACACCTCCC-3′ |
| | Reverse | 5′-GTTCAACTCCAAGTCACTCATG-3′ |
| **Ccl5** | Forward | 5′-CCCACGTCAAGGAGTATTTCT-3′ |
| | Reverse | 5′-ACCCTCTATCCTAGCTCATCTC-3′ |
| **Cd83** | Forward | 5′-CGCAGCTCTCCTATGCAGTG-3′ |
| | Reverse | 5′-GTGTTTTGGATCGTCAGGGAATA-3′ |
| **Clec7a** | Forward | 5′-GAATCCTGTGCTTTGTGGTAG-3′ |
| | Reverse | 5′-GGAGCCACCTTCTCATCTAAAG-3′ |
| **Entpd1** | Forward | 5′-ACCGAACTGATACCAACATCC-3′ |
| | Reverse | 5′-CCTTCCTCTTGTCCAGTGATG-3′ |
| **Ifng** | Forward | 5′-GCTTTGCAGCTCTTCCTCAT-3′ |
| | Reverse | 5′-GTCACCATCCTTTTGCCAGT-3′ |
| **Il6** | Forward | 5′-ACAAAGCCAGAGTCCTTCAGAG-3′ |
| | Reverse | 5′-GAGCATTGGAAATTGGGGTAGG-3′ |
| **Il12a** | Forward | 5′-AACAGGGTGATGGGCTATC-3′ |
| | Reverse | 5′-TGAGATGTGATGGGAGAACAG-3′ |
| **Il23a** | Forward | 5′-GCAACTTCACACCTCCCTAC-3′ |
| | Reverse | 5′-CGAAGGATCTTGGAACGGAG-3′ |
| **Lpl** | Forward | 5′-TAGACAACGTCCACCTCTTAG-3′ |
| | Reverse | 5′-ACATCTACAAAATCAGCGTCATC-3′ |
| **Mmp9** | Forward | 5′-GCTGACTACGATAAGGACGGCA-3′ |
| | Reverse | 5′-TAGTGGTGCAGGCAGAGTAGGA-3′ |
| **Msr1** | Forward | 5′-AGGTGTTAAAGGTGATCGGG-3′ |
| | Reverse | 5′-ATCTTGATCCGCCTACACTC-3′ |
| **Rpl4** | Forward | 5′-GCTGAACCCTTACGCCAAGA-3′ |
| | Reverse | 5′-TCTCGGATTTGGTTGCCAGT-3′ |
| **Siglech** | Forward | 5′-GCAAGAGATCCCATACGAAGAG-3′ |
| | Reverse | 5′-AGTCCAGTTGGCACCATC-3′ |
| Spp1 | Forward | 5′-ACTTTCACTCCAATCGTCCC-3′ |
| | Reverse | 5′-GTCCTCATCTGTGGCATCAG-3′ |
| **Tbx21** | Forward | 5′-AGCAAGGACGGCGAATGTT-3′ |
| | Reverse | 5′-GGGTGGACATATAAGCGGTTC-3′ |
| **Tmem119** | Forward | 5′-GTGTCTAACAGGCCCCAGAA-3′ |
| | Reverse | 5′-AGCCACGTGGTATCAAGGAG-3′ |
| **Tnf** | Forward | 5′-GTGATCGGTCCCCAAAGGG-3′ |
| | Reverse | 5′-CCAGCTGCTCCTCCACTTG-3′ |
| **Trem2** | Forward | 5′-GGAACCGTCACCATCACTC-3′ |
| | Reverse | 5′-CCAGCATCTTGGTCATCTAGAG-3′ |
| **CD83 (human)** | Forward | 5′-TGCTGCTGGCTCTGGTTATT-3′ |
| | Reverse | 5′-TGTGAGGAGTCACTAGCCCT-3′ |
| **RPLP0 (human)** | Forward | 5′GCCATTGCCCCATGTGAAGT3′ |
| | Reverse | 5′AGCTGCACATCACTCAGGATT3′ |

in 0.3% Triton-X in PBS, containing 5% normal donkey serum. The sections were incubated overnight at 4 °C with the primary antibodies (mouse anti-CD83, HB15a, Beckman Coulter, and polyclonal goat anti-Iba1, NB100–1028, Novus Biologicals, both diluted 1:250), followed by staining with secondary antibodies (donkey-anti-mouse-AlexaFluor488,

Jackson Immunoresearch, 1:250, and donkey-anti-goat-AlexaFluor568, Invitrogen, 1:250) and DAPI for 2 h at RT. The stained sections were subsequently mounted onto glass slides and mounted using the aqueous mounting medium Faramount (Agilent Dako, UK) under glass coverslips.

## Two-photon-microscopy

Brains and spinal cords were isolated at day 16 after EAE induction from perfused mice and cut into 2 mm thin sections. Tissues were washed three times in 1 ml PBS and subsequently fixed and permeabilized in 4% PFA/0.1% Triton-X-100 in PBS on a shaker (1 h, 800 rpm, 4 °C). Afterward, sections were incubated with blocking buffer (1% BSA/0.1% Triton-X-200 in PBS) on a shaker (1 h, 800 rpm, 4 °C). Samples were washed three times in PBS and primary antibodies against P2YR12 (Biolegend, S16007D) diluted in blocking buffer were added for 48 h. Samples were washed four times in PBS for 30 min on a shaker and incubated with the appropriate secondary antibody on a shaker (48 h, 800 rpm, 4 °C). Confocal imaging of brain regions was performed on a Zeiss LSM880 NLO Two-Photon microscope, equipped with a 680–1300 nm tunable and a fixed 1040 nm two-photon laser from Newport SpectraPhysics. Two-photon images of brain regions were acquired with a 20x W-Plan Apochromatic objective water dipping lens. The fluorophores AF633 (for P2YR12) and eGFP were excited at 940 nm, and specific emission was detected at 500–550 nm and 640–710 nm.

## Statistical analysis and reproducibility

All experiments, where only representative pictures are shown, were performed successfully at least two times. Statistics (except those shown for the evaluation of scRNA-Seq data) were performed with GraphPad Prism 9.3.1 using the Mann–Whitney U-test (two-tailed) for non-parametric data or one-way ANOVA. Data are presented as mean values, including standard error of the mean (SEM). $p$-values of $^*p < 0.05$, $^{**}p < 0.01$, $^{***}p < 0.001$ and $^{****}p < 0.0001$ were considered statistically significant.

## Reporting summary

Further information on research design is available in the Nature Portfolio Reporting Summary linked to this article.

## Data availability

The RNA-seq datasets reported in this study can be found at Gene Expression Omnibus under the accession code GSE230753. RNA-Seq data from the RR-EAE study are accessible via GSE214709[29]. Source data are provided with this paper.

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

## Acknowledgements

We would like to thank all members of the Steinkasserer Group, as well as the staff of the central animal facility, the FACS core facility, and the core facility NGS-sequencing for expert technical help. We further thank the group of Steffen Jung (Weizmann Institute of Science) for generously providing their sequencing data from RR-EAE animals[29], which greatly helped to substantiate our data. The authors are grateful for patients donating tissue for research. This work was supported by the Deutsche Forschungsgemeinschaft (DFG) within grants SFB1181 project B03 and STE432/15-1 (to A.S.), RA2506/6-1 and SFB1181 project C06 (to A.R.); by the European Research Council (853508 BARRIER BREAK) to A.R., by the Bundesministerium für Bildung und Forschung (MASCARA to A.R.); by the Interdisciplinary Center of Clinical Studies (IZKF) at the University Hospital of the FAU Erlangen-Nuremberg to A.S. (grant A89). A.B.W. was supported by the Else Kröner-Fresenius-Stiftung (grant 2020_EKEA.81) and the ELAN Fond of the IZKF at the University Hospital of the FAU Erlangen-Nuremberg (grant P062). K.P.M. was funded by the Bavarian Equal Opportunities Sponsorship–Realization Equal Opportunities for Women in Research and Teaching, and the ELAN Fond of the IZKF at the University Hospital of the FAU Erlangen-Nuremberg (grant P105). V.P. was supported by an intramural funding program of the Faculty of Medicine, University of Tübingen (Fortüne 2707-0-0). Z.H. was supported by the DFG (grant A02 within the CRC/TRR167).

## Author contributions

A.S. and A.B.W. conceived the project and designed the experiments; P.S., K.P.M., C.D., C.K., and A.B.W. performed the experiments; K.P.M. critically helped with two-photon-imaging; V.P. performed experiments on human iPSC-derived human microglia and staining of human brain tissue; S.R. supervised the generation of scRNA-seq libraries, interpreted the data and generated the figures. H.M. performed the analysis of the scRNA-seq data, interpreted the data and generated the figures; Z.H. provided, assessed, and visualized relevant data on the RR-EAE model; M.L. provided and maintained neonatal microglial cultures; P.L. and A.B.W. evaluated the results and wrote the manuscript. D.R. advised on data interpretation; D.K.V., A.R., and A.S. revised the manuscript; A.S. and A.B.W. supervised the project and acquired funding.

## Funding

## Competing interests

The authors declare no competing interests.
