## [Peer Review File · Nature Communications]

REVIEWER COMMENTS

Reviewer #1 (Remarks to the Author):

Langguth et al have studied the role of CD83 in microglia under homeostatic and pathologic conditions. This glycoprotein has been previously shown by the same group to exert regulatory effect on DCs, and is also implicated in control of Treg generation. Here they have used a CD83 promoter driven GFP reporter mouse to show that great majority of cells expressing CD83 in CNS are microglia, with increased signal in caudal regions and in spinal cord. Expression was increased on microglia in EAE-inflamed CNS especially on CD11c+/MHCII+ cells, and was maintained in the recovery and remission phases when CD11c declined. Expression of CD83 was increased by TGFb and IL-4 in vitro whereas TNFa and LPS reduced it – comparing to Tmem119 and Msr1 suggested a broadly similar pattern to regulation of homeostatic gene transcripts. They then deleted CD83 from microglia using a CX3CR1-CreERT2 approach and showed that DAM-associated Trem2 and Lpl were reduced. CD83-deficient microglia (CD83ΔMG) could phagocytose myelin but did not induce Lpl and they showed increased Ccl2 expression. CD83ΔMG mice showed more severe EAE with infiltration of myeloid cells (monocytes and MDCs) and increased proportion of CD11c+/MHCII+ microglia (although overall microglial numbers were reduced). CD83ΔMG microglia expressed elevated levels of message for CCL2, CCL4 and CCL5, and reduced levels of Tmem119 and Trem2 and Lpl. Frequency of Tregs in EAE CNS was reduced and IFNg-producing Th1 cells increased in CD83ΔMG mice. Th17 appeared unaffected. TNFa and MMP9 transcripts were dramatically elevated. They conclude that CD83 on microglia acts as a gatekeeper of cellular activation in neuroinflammation, and is connected with a pro-resolving state of microglia, associated to appropriate removal of myelin debris and regulation of inflammation.

The topic and the data are of interest. The findings complement this group's previous studies of CD83 on DC and T cells, including that EAE was dramatically aggravated in CD83ΔDC mice while resolution of inflammation was strongly reduced. They now extend this to microglia.

There are some over-interpretations which are understandable but might be modulated in revision.

- It is not clear that 'prolonged CD83 expression is tightly connected with a pro-resolving state of microglia' has actually been demonstrated, although it can certainly be speculated.
- The statement on p 8 that aggravated EAE was driven by a heightened influx of monocytic cells is based on circumstantial evidence
- Similarly, the 'derailed response to damaged axons' is a bit speculative.
- The RR-EAE model is not described.
- Data for a dynamic reaction of cultured microglia to environmental changes (p 5) are not shown, in Fig 2A or anywhere else.

Para 1 of Results, the markers used for B, T, NK cells and monocytes should be described

Fig 1E –all mice are from Day 16 EAE - were these all the same grades?

On p 7, relative frequencies of microglia, monocytes and MDCs are given, but absolute numbers are needed, since percentages can change without change in numbers

On p 7, Cd83 expression is described for MDCs, what about DCs? Monocytes? Neutrophils?

P 9 – increased microglial expression of Cd83 in white matter regions should be shown by immunofluorescence

P 12 , last paragraph – should call Fig 7, not Fig 6

Methods – please give information on proportions of male versus female mice in these experiments

Reviewer #2 (Remarks to the Author):

Langguth and colleagues studied the involvement of CD83 in the phenotypic alteration of microglia and how CD83 deficiency gives an impact on disease progression, focusing on autoimmune neuroinflammation. They found that CD83 is upregulated not only in microglia, but other immune cells such as monocyte during disease, and deficiency of CD83 in microglia enhances proinflammatory responses, which exacerbates the disease. In general, this study is interesting and provides novel aspects of how microglial phenotypes are controlled through CD83. The paper is clearly written, and the abstract is appropriate. However, while I appreciate the broadness of the study, the paper is lacking in-depth analysis of how CD83 controls microglial phenotypes. Thus, I suggest that the authors drill down the mechanism of CD83-mediated phenotypic regulation in microglia to increase the significance and the novelty of the study. For instance, profiling of CD83-deficient microglia by RNA-seq or similar would be helpful. In addition, recent studies have shown the diversity of microglia in health and disease, suggesting distinct roles of microglia subsets/substates in each context. Indeed, microglia have been shown to become more diverse during the course of EAE (PMID: 30679343). So, this study would benefit from the aspect of heterogeneity of microglia that may be controlled by CD83. How does CD83 expression contribute to the diversification of microglia? Or CD83-expressing population arises during disease?

Additional points

1. The authors showed that TGF- β signaling contributes to CD83 expression in cultured microglia. However, TGF- β expression is shown to be downregulated in the disease context. How do the authors explain this discrepancy?

2. As the authors mentioned in the introduction, Cd83 expression is also expressed in border-associated macrophages. Considering that Cx3cr1-CreER is targeting both microglia and the macrophages (PMID: 32541832), the possible contribution of CD83 in border-associated macrophages to disease progression cannot be excluded.

Minor points

1. The gating strategies to isolate each cell-type or population should be included somewhere in the paper.

2. The images shown in Supplementary Figure S1B are unclear and should be replaced.

Reviewer #3 (Remarks to the Author):

Provided is a characterization of the role of CD83 in mouse microglia. To this, CD83-eGFP reporter and inducible CD83 Δ MG knockout mice were applied. The work is interesting, novel, innovative, and well presented. I have a couple of specific comments:

1. The authors repeatedly talk about microglia subsets. The existence of microglia subsets is not generally accepted (PMID 32579115). Notably, also this manuscript – based on CD83 – does not provide arguments that would justify a subdivision of microglia into subsets. So,

please, either rephrase or support statements in this direction.

2. Across the entire paper, I noted that relatively subtle changes are strongly emphasized, and I got the feeling that significance is sometimes mixed up with effect size. The authors should be more cautious and consider rephrasing some of their statements, eg:

a. Fig 1D, leading to the conclusion that CD83 expression shows a 'distinctive regional pattern'. In fact, all microglia in the brain locations tested are CD83+, and expression levels vary by at most 50%. This suggests to me a 'similar regional pattern'.

b. Fig 3D and S2D, reporting 'disturbed microglia homeostasis' based on changes in the expression of Trem2 and Apoe in the order of 10%. To me, it seems that more arguments are needed to indicate that the homeostasis of microglia is affected.

c. Fig 3F, claiming an 'aberrant response to myelin debris' based on altered expression of two genes (Lpl and Ccl2), of which the first one is already regulated in the absence of myelin uptake. How about morphological changes and the expression of other genes related to myelin uptake (Gpnmb, Chit1, Ccl18)?

3. The finding that anti-inflammatory cytokines up-regulate and pro-inflammatory cytokines downregulate CD83 is intriguing but also seems contradictory with observations that CD83 is upregulated in microglia in EAE and MS (PMID 29312322). I noted the higher expression during EAE remission but still find it difficult to get a clear picture. In my view, this part of the study requires the addition of data on the expression of CD83 in human microglia to increase relevance and robustness of the findings, which are now all based on experimental systems (reporter mice, in vitro cultures, EAE). Notably, Masuda et al found CD83 expression only in a small fraction of homeostatic human microglia (PMID 30760929). It seems to me that monoclonal antibodies are available that could be applied to test CD83 expression on freshly isolated human microglia by flow cytometry. Further, data mining could help to understand the regulation of microglial CD83 in multiple sclerosis.

4. Absence of CD83 expression worsens disease activity in the EAE model. This is attributed to microglia as they do not restore CD83 expression over time due to their long life-span. Monocytes and cells derived from monocytes restore CD83 expression several weeks after tamoxifen treatment. This sounds fine, yet why only peripheral cells expressing CD83 (monocytes, MDCs, moDCs) entered the EAE brains in larger numbers? Why PMN- and T-cell counts stay normal (these data should be shown) despite T-cell-attracting cytokines were elevated as well? Notably, Treg-cell infiltration was even reduced. But Tregs cells need CD83 for their development as shown by the authors themselves (PMID 29875316). With this in mind, I think that relating the effects seen in the EAE model to (lack of) microglial CD83 needs further confirmation, e.g. the testing of bone marrow chimeras.

Minor points

- Fig 1E/F – Please, check legends, which read different while the panels are related
- Figs 3B and S3A show identical items

Reviewer #1:

Langguth et al have studied the role of CD83 in microglia under homeostatic and pathologic conditions. This glycoprotein has been previously shown by the same group to exert regulatory effect on DCs, and is also implicated in control of Treg generation. Here they have used a CD83 promoter driven GFP reporter mouse to show that great majority of cells expressing CD83 in CNS are microglia, with increased signal in caudal regions and in spinal cord. Expression was increased on microglia in EAE-inflamed CNS especially on CD11c+/MHCII+ cells, and was maintained in the recovery and remission phases when CD11c declined. Expression of CD83 was increased by TGF β and IL-4 in vitro whereas TNF α and LPS reduced it – comparing to Tmem119 and Msr1 suggested a broadly similar pattern to regulation of homeostatic gene transcripts. They then deleted CD83 from microglia using a CX3CR1-CreERT2 approach and showed that DAM-associated Trem2 and Lpl were reduced. CD83-deficient microglia (CD83 Δ MG) could phagocytose myelin but did not induce Lpl and they showed increased Ccl2 expression. CD83 Δ MG mice showed more severe EAE with infiltration of myeloid cells (monocytes and MDCs) and increased proportion of CD11c+/MHCII+ microglia (although overall microglial numbers were reduced). CD83 Δ MG microglia expressed elevated levels of message for CCL2, CCL4 and CCL5, and reduced levels of Tmem119 and Trem2 and Lpl. Frequency of Tregs in EAE CNS was reduced and IFN γ -producing Th1 cells increased in CD83 Δ MG mice. Th17 appeared unaffected. TNF α and MMP9 transcripts were dramatically elevated. They conclude that CD83 on microglia acts as a gatekeeper of cellular activation in neuroinflammation, and is connected with a pro-resolving state of microglia, associated to appropriate removal of myelin debris and regulation of inflammation.

The topic and the data are of interest. The findings complement this group's previous studies of CD83 on DC and T cells, including that EAE was dramatically aggravated in CD83 Δ DC mice while resolution of inflammation was strongly reduced. They now extend this to microglia. There are some over-interpretations which are understandable but might be modulated in revision.

- It is not clear that 'prolonged CD83 expression is tightly connected with a pro-resolving state of microglia' has actually been demonstrated, although it can certainly be speculated.
- The statement on p 8 that aggravated EAE was driven by a heightened influx of monocytic cells is based on circumstantial evidence
- Similarly, the 'derailed response to damaged axons' is a bit speculative.

Response: *We thank the reviewer for taking interest in our presented data and for summarizing our findings so splendidly. Since also Reviewer#3 called for a more careful interpretation of the data, we re-evaluated our results and modified the text in a way that it does not overstate the drawn conclusions.*

- The RR-EAE model is not described.

Response: *By the time we first submitted this manuscript, the information on the RR-EAE model was included in a not yet published study by the group of Steffen Jung. Since we only obtained data from them, the methodology is not fully described in this manuscript but can be accessed by the respective study of the Jung Lab (see Haimon et al., 2022, Nat. Immunol).*

- Data for a dynamic reaction of cultured microglia to environmental changes (p 5) are not shown, in Fig 2A or anywhere else.

Response: *We included now two videos showing the dynamic surveillance of the environment (Supplementary Movie 1, white arrow) and the active uptake of pHrodo[®]-labelled myelin particles (Supplementary Movie 2).*

Para 1 of Results, the markers used for B, T, NK cells and monocytes should be described

Response: *This is a valid point and we included a full gating strategy in the supplement (see Supplementary Fig. 1).*

Fig 1E –all mice are from Day 16 EAE - were these all the same grades?

Response: *Animals in these experiments had a disease score between 3 and 4 (clear paralytic symptoms).*

On p 7, relative frequencies of microglia, monocytes and MDCs are given, but absolute numbers are needed, since percentages can change without change in numbers

Response: *We agree with the reviewer on the point that percentages can change without affecting total numbers. However, we chose this representation to demonstrate an increased influx of peripheral immune cells into the CNS of CD83 cKO mice. In healthy animals, the majority (>90%) of isolated CD45+ cells are microglia and thus, the lower the percentage is, the greater is the contribution of peripheral infiltrating cells. By showing the percentages we want to illustrate that the cKO animals experience a more severe infiltration of (in this case) monocytic cells, which contributes to worsened disease outcome.*

On p 7, Cd83 expression is described for MDCs, what about DCs? Monocytes? Neutrophils?

Response: *We included the analysis of MDCs in Supplementary Fig. 4 to demonstrate that CD83 is only absent on resident microglial cells but not on infiltrating effector cells in tamoxifen-treated cKO mice. As shown in Fig. 1 and Supplementary Fig. 1, neutrophils do not express CD83 during EAE, whereas monocytes can gain CD83 during transition to MDCs (i.e. acquisition of CD11c and MHC-II). Further, we did not include DCs in these analyses because we did not observe an effect of tamoxifen-mediated recombination on splenic DCs after seven weeks, when the EAE experiments commenced (see also Supplementary Fig. 3).*

P 9 – increased microglial expression of Cd83 in white matter regions should be shown by immunofluorescence

Response: *On the recommendation of this reviewer, we performed immunofluorescent stainings of human brain tissue that included grey and white matter (see Fig. 1e). By that, we confirmed the published RNA data from sorted human microglia.*

P 12 , last paragraph – should call Fig 7, not Fig 6

Response: *We thank the reviewer for noticing this mistake. We corrected the figure labeling in the revised version. Due to the inclusion of scRNA-Seq experiments, the figure numbering has globally changed, which is why this figure is now referred to as Fig.8.*

Methods – please give information on proportions of male versus female mice in these experiments

Response: *We included a statement of male/female proportion into the methods section.*

Reviewer #2 (Remarks to the Author):

Langguth and colleagues studied the involvement of CD83 in the phenotypic alteration of microglia and how CD83 deficiency gives an impact on disease progression, focusing on autoimmune neuroinflammation. They found that CD83 is upregulated not only in microglia, but other immune cells such as monocyte during disease, and deficiency of CD83 in microglia enhances proinflammatory responses, which exacerbates the disease. In general, this study is interesting and provides novel aspects of how microglial phenotypes are controlled through CD83. The paper is clearly written, and the abstract is appropriate. However, while I appreciate the broadness of the study, the paper is lacking in-depth analysis of how CD83 controls microglial phenotypes. Thus, I suggest that the authors drill

down the mechanism of CD83-mediated phenotypic regulation in microglia to increase the significance and the novelty of the study. For instance, profiling of CD83-deficient microglia by RNA-seq or similar would be helpful.

Response: *We are very grateful for this comment, based on which we have performed scRNA-Seq on both healthy and EAE-experienced microglia from the CD83cKO mice. By this, it got very clear that CD83-deletion only mildly affects microglia under homeostatic conditions. As we have observed in previous studies on dendritic cells and macrophages (Wild et al. 2019, Peckert-Maier et al. 2023), CD83-deletion rather affects the cellular function after a certain type of activation – in the case of CD83cKO microglia changes got overt in EAE-samples. Here, we demonstrated that CD83-deficient microglia got over-activated compared to their wt counterparts, and that CD83 expression on microglia is correlating with a dampened response towards external stimuli.*

In addition, recent studies have shown the diversity of microglia in health and disease, suggesting distinct roles of microglia subsets/substates in each context. Indeed, microglia have been shown to become more diverse during the course of EAE (PMID: 30679343). So, this study would benefit from the aspect of heterogeneity of microglia that may be controlled by CD83. How does CD83 expression contribute to the diversification of microglia? Or CD83-expressing population arises during disease?

Additional points

1. The authors showed that TGF- β signaling contributes to CD83 expression in cultured microglia. However, TGF- β expression is shown to be downregulated in the disease context. How do the authors explain this discrepancy?

Response: *We thank the reviewer for this deliberate question. In our opinion, this discrepancy relies on the difficulties of microglia ex vivo analyses. As shown by Bohlen et al. 2017, TGF- β is an important factor when cultivating microglia in vitro. With our culture system, we aimed to demonstrate that Cd83-expression is as dependent on TGF- β as microglia homeostatic genes like Tmem119. We also showed that Cd83 can be induced by IL-4 and it is conceivable that a complex neuro-inflammatory milieu also contains factors that induce its expression. Therefore, down-modulation of TGF- β during disease does not contradict our findings on Cd83 being TGF- β -dependent in vitro.*

2. As the authors mentioned in the introduction, Cd83 expression is also expressed in border-associated macrophages. Considering that Cx3cr1-CreER is targeting both microglia and the macrophages (PMID: 32541832), the possible contribution of CD83 in border-associated macrophages to disease progression cannot be excluded.

Response: *We thank the reviewer for pointing out this very important fact. Indeed, targeting of BAMs cannot be fully excluded by the tamoxifen-inducible approach. Although absence of CD83 on BAMs might influence the disease course in the EAE model, we always included anti-P2RY12 in the staining-mixes for microglia (for FACS-analyses or sorting for gene expression analyses). As this marker is absent on BAMs (<https://brainimmuneatlas.org/tsne-aggrfull.php>), we are confident that CD83 expression by BAMs does not confound the validity of our study regarding the function of CD83 in microglia. Additionally, we excluded possibly contaminating macrophages from our scRNA-Seq analysis (see Supplementary Fig. 5b), which corroborates the notion that observed changes in the phenotype (e.g. higher activation status) is really an intrinsic factor of CD83-deficient microglia.*

Minor points

1. The gating strategies to isolate each cell-type or population should be included somewhere in the paper.

Response: *This is a valid point and we included a full gating strategy in the supplement (see Supplementary Fig. 1).*

2. The images shown in Supplementary Figure S1B are unclear and should be replaced.

Response: *We like to apologize for the possibly unclear depiction in this figure. However, the Two-Photon-Approach we chose in this experiment offers a deeper tissue penetration at the expense of resolution. We only included this picture to further visualize and corroborate the findings from our flow cytometry data. If this reviewer still feels that we should remove the figure, we certainly will defer to that opinion.*

Reviewer #3 (Remarks to the Author):

Provided is a characterization of the role of CD83 in mouse microglia. To this, CD83-eGFP reporter and inducible CD83 Δ MG knockout mice were applied. The work is interesting, novel, innovative, and well presented. I have a couple of specific comments:

1. The authors repeatedly talk about microglia subsets. The existence of microglia subsets is not generally accepted (PMID 32579115). Notably, also this manuscript – based on CD83 – does not provide arguments that would justify a subdivision of microglia into subsets. So, please, either rephrase or support statements in this direction.

Response: *We thank the reviewer for making this important point. We have now performed scRNA-Seq and indeed observed a rather uniform expression pattern of CD83 in microglia. However, in line with human data, there is a specific cluster (Ccl4⁺) where we observed particularly high Cd83 expression. Also, we detected unevenly distributed CD83 in tissue sections of human brains. We are well aware that these data might not justify a general subdivision of microglia into subsets but they clearly show an evolutionary conserved association of CD83 with microglial cells that exhibit a certain gene expression pattern. Nonetheless, we re-wrote the respective sections and abstained from the term “microglia subset” to avoid further confusion.*

2. Across the entire paper, I noted that relatively subtle changes are strongly emphasized, and I got the feeling that significance is sometimes mixed up with effect size. The authors should be more cautious and consider rephrasing some of their statements, eg:

a. Fig 1D, leading to the conclusion that CD83 expression shows a ‘distinctive regional pattern’. In fact, all microglia in the brain locations tested are CD83+, and expression levels vary by at most 50%. This suggests to me a ‘similar regional pattern’.

Response: *We agree with the reviewer on the point that CD83 expression does not vary greatly. However, we detected significantly elevated expression of CD83 in the spinal cord, which was already demonstrated in previous studies. Thus, we wanted to express that CD83 is not uniformly expressed throughout the brain but shows differences in distinct regions. Since this reviewer also pointed out a difference between significance and effect size, we calculated Cohen’s d for this experiment. We obtained values of $d > 2$ ($d_{\text{Ctx/STEM}}=2.46$; $d_{\text{Ctx/Cbm}}=3.82$; $d_{\text{Ctx/SC}}=5.33$) for all comparisons except Ctx/Hip, showing strong effects.*

b. Fig 3D and S2D, reporting ‘disturbed microglia homeostasis’ based on changes in the expression of Trem2 and Apoe in the order of 10%. To me, it seems that more arguments are needed to indicate that the homeostasis of microglia is affected.

Response: *We totally consent with the reviewer on the valid point that the term ‘disturbed microglia homeostasis’ might have overstated the results. Although we also showed reduction in expression of CD39 and Siglec H as well as Lpl in the steady state, which implies changes in the “normal” function of these cells, the more important impact of CD83-deletion is seen under inflammatory conditions. We therefore rephrased the respective sections to avoid confusion and overstating the results.*

c. Fig 3F, claiming an ‘aberrant response to myelin debris’ based on altered expression of two genes (Lpl and Ccl2), of which the first one is already regulated in the absence of myelin uptake. How about

morphological changes and the expression of other genes related to myelin uptake (Gpnmb, Chit1, Ccl18)?

Response: *These are very important concerns. Regarding the impaired upregulation of Lpl on CD83cKO cells, we agree that this may spark confusion and removed the graph. We checked several other genes (e.g. Gpnmb, see Fig. R1) but did not detect any differences between both groups.*

Figure R1: Gene expression analysis of Gpnmb in microglial cultures from CD83^{wt^{Cre}} or CD83^{ΔMG} either untreated (NTC) or after stimulation with myelin debris. Gene expression is depicted as x-fold induction over NTC (n=9, pooled from four independent experiments)

We included the 'aberrant response to myelin' in the data because we consistently observed reduced levels of Trem2 and Lpl in the cKO cells, which we think is an important factor contributing to the exaggerated activation in response to neuroinflammation. We are aware that in vitro stimulation of microglia can only be a surrogate for the mechanisms happening during EAE but we hope to have convinced this reviewer that the results support the overall view on the role of CD83 in microglia.

3. The finding that anti-inflammatory cytokines up-regulate and pro-inflammatory cytokines downregulate CD83 is intriguing but also seems contradictory with observations that CD83 is upregulated in microglia in EAE and MS (PMID 29312322). I noted the higher expression during EAE remission but still find it difficult to get a clear picture. In my view, this part of the study requires the addition of data on the expression of CD83 in human microglia to increase relevance and robustness of the findings, which are now all based on experimental systems (reporter mice, in vitro cultures, EAE). Notably, Masuda et al found CD83 expression only in a small fraction of homeostatic human microglia (PMID 30760929). It seems to me that monoclonal antibodies are available that could be applied to test CD83 expression on freshly isolated human microglia by flow cytometry. Further, data mining could help to understand the regulation of microglial CD83 in multiple sclerosis.

Response: *We are aware that the difference between neuroinflammation and our in vitro system could cause confusion. In our opinion, the activation of microglia during EAE/MS is multi-dimensional (cytokines, dead cells, debris) and it might be rather hard to compare this situation with isolated stimuli in the cell culture. We aimed to show that Cd83-expression is NOT only tied to cellular activation alone (as it is frequently assumed) but rather multi-faceted. Following this reviewer's advice, we also included data on stimulated human microglia, and observed a similar pattern. Additionally, we provided evidence that CD83 expression is clearly associated with microglia of the white matter (in humans) and it also seemed that the CD83-positive fraction had a different morphology. Unfortunately, we did not have access to acutely isolated human microglia but are confident that the presented data satisfyingly clear out the raised issues.*

4. Absence of CD83 expression worsens disease activity in the EAE model. This is attributed to microglia as they do not restore CD83 expression over time due to their long life-span. Monocytes and cells derived from monocytes restore CD83 expression several weeks after tamoxifen treatment. This sounds fine, yet why only peripheral cells expressing CD83 (monocytes, MDCs, moDCs) entered the

EAE brains in larger numbers? Why PMN- and T-cell counts stay normal (these data should be shown) despite T-cell-attracting cytokines were elevated as well?

Notably, Treg-cell infiltration was even reduced. But Tregs cells need CD83 for their development as shown by the authors themselves (PMID 29875316).

Response: *We highly appreciate the reviewer's detailed knowledge on our previous data. However, we have to point out that the mentioned study demonstrates that Treg-intrinsic expression is needed for their stability and differentiation. The present study focuses on the fact that CD83-depletion in microglia can impact on the balance between pathogenic T-helper cells and protective Tregs during the course of EAE. Regarding the imbalance on PMN and T cells, we sincerely apologize for any inconvenience as this assumption was based on a preliminary data set, which unfortunately had not been updated. After rectifying this omission, we indeed observed an elevated infiltration of lymphocytes into the CNS, which is in line with the elevated chemokine levels (see new Supplementary Fig. 4c). PMN numbers might show no difference because the CNS was analyzed at the peak of disease, when neutrophil infiltration does not play the main part anymore.*

With this in mind, I think that relating the effects seen in the EAE model to (lack of) microglial CD83 needs further confirmation, e.g. the testing of bone marrow chimeras.

Response: *We genuinely hope that we have convinced the reviewer of the validity of our results by addressing all raised questions above. Despite minor limitations regarding specificity, the tamoxifen-induced knock-out via the CX3CR1-CreERT-deleter is a well-established and widely accepted system to study effects on microglial biology under different conditions. Thus, we do not feel that addition of experiments with bone-marrow chimera would add new insights to our study.*

Minor points

- Fig 1E/F – Please, check legends, which read different while the panels are related
- Figs 3B and S3A show identical items

Response: *We cordially thank the reviewer for pointing this out. We rephrased the legend of Fig. 1 and implemented the correct graph in the new Fig. 4b.*

REVIEWERS' COMMENTS

Reviewer #1 (Remarks to the Author):

The authors have responded to my comments and those of the other reviewers with revisions and new data that make this a much better paper which should now be published.

Reviewer #2 (Remarks to the Author):

The authors addressed most of the reviewer's concerns and the modifications made including the new data have strengthened the manuscript and its conclusions. However, some points need to be further addressed.

1. As for the CD83 expression, without in vivo validation the data shown in Fig.3 is weak and even confusing since the cultured conditions don't reflect in vivo situations (Tgfb and IL4 are induced during EAE?). The authors may move them to the Supplementary Figs.

2. The reviewer is fully aware that some of the experiments, such as FACS-analyses or gene expression analyses with sorted cells, have been nicely conducted with Cx3cr1CreER-mediated CD83-KO mice, and the data are convincing. Still, without microglia-specific Cre-deletor (e.g. Tmem119-CreER, Hexb-CreER), the possible contribution of CD83 in border-associated macrophages to the disease progression (Fig.4e), proportion of immune cells (Fig.4g), and T cell balance (Fig.7) cannot be excluded, which would be a technical limitation. The authors should mention this point in the manuscript.

Minor

Line 239: WT mice mean CD83wtCre? If so, the authors should rephrase it. In addition, I do not see the point of the sentence "In line with data from...".

Reviewer #3 (Remarks to the Author):

I see my comments are adequately addressed.

Reviewer #1 (Remarks to the Author):

The authors have responded to my comments and those of the other reviewers with revisions and new data that make this a much better paper which should now be published.

Response: *We cordially thank this reviewer for her/his advice, which helped us to improve our manuscript tremendously.*

Reviewer #2 (Remarks to the Author):

The authors addressed most of the reviewer's concerns and the modifications made including the new data have strengthened the manuscript and its conclusions. However, some points need to be further addressed.

1. As for the CD83 expression, without in vivo validation the data shown in Fig.3 is weak and even confusing since the cultured conditions don't reflect in vivo situations (Tgfb and IL4 are induced during EAE?). The authors may move them to the Supplementary Figs.

Response: *First of all, we are grateful for all offered advice to strengthen our study. Nonetheless, we deeply regret that we could not convince this reviewer of the cogency of the results from our in vitro systems. In our opinion, they demonstrate a similar regulation of CD83 in murine and human microglia, which is a novel finding. However, we accept that these data might be confusing in the context of the whole paper focusing on neuroinflammation and therefore, we moved the entire figure to the supplement (now part of Supplementary Fig. 2). We also shortened the respective section of the results accordingly, and hope that this modified version will meet with the reviewer's approval.*

2. The reviewer is fully aware that some of the experiments, such as FACS-analyses or gene expression analyses with sorted cells, have been nicely conducted with Cx3cr1CreER-mediated CD83-KO mice, and the data are convincing. Still, without microglia-specific Cre-deletor (e.g. Tmem119-CreER, Hexb-CreER), the possible contribution of CD83 in border-associated macrophages to the disease progression (Fig.4e), proportion of immune cells (Fig.4g), and T cell balance (Fig.7) cannot be excluded, which would be a technical limitation. The authors should mention this point in the manuscript.

Response: *We agree with this reviewer on the point that CD83 is also expressed in BAMs, which could impact on the disease course. We therefore added a respective statement in the discussion to point out this limitation.*

Minor

Line 239: WT mice mean CD83wtCre? If so, the authors should rephrase it. In addition, I do not see the point of the sentence "In line with data from...".

Response: *We apologize for this inconsistency and changed "WT" to "CD83wt^{Cre}". We also rephrased the entire sentence as it was obviously not worded carefully enough.*

Reviewer #3 (Remarks to the Author):

I see my comments are adequately addressed.

Response: *We wholeheartedly thank this reviewer for all of her/his suggestions, which greatly improved the validity of our study.*